# Reconfiguration of the reductive TCA cycle enables high-level succinic acid production by *Yarrowia lipolytica*

Zhiyong Cui[1], Yutao Zhong[1], Zhijie Sun[2], Zhennan Jiang[1], Jingyu Deng[1], Qian Wang[3], Jens Nielsen [4,5], Jin Hou [1] ✉ & Qingsheng Qi [1] ✉

Succinic acid (SA) is an important C4-dicarboxylic acid. Microbial production of SA at low pH results in low purification costs and hence good overall process economics. However, redox imbalances limited SA biosynthesis from glucose via the reductive tricarboxylic acid (TCA) cycle in yeast. Here, we engineer the strictly aerobic yeast *Yarrowia lipolytica* for efficient SA production without pH control. Introduction of the reductive TCA cycle into the cytosol of a succinate dehydrogenase-disrupted yeast strain causes arrested cell growth. Although adaptive laboratory evolution restores cell growth, limited NADH supply restricts SA production. Reconfiguration of the reductive SA biosynthesis pathway in the mitochondria through coupling the oxidative and reductive TCA cycle for NADH regeneration results in improved SA production. In pilot-scale fermentation, the engineered strain produces 111.9 g/L SA with a yield of 0.79 g/g glucose within 62 h. This study paves the way for industrial production of biobased SA.

Succinic acid (SA) is a promising platform chemical as it can be used for production of 1,4-butanediol, γ-butyrolactone, tetrahydrofuran, and polybutylene succinate (PBS)[1,2]. Because of its wide applications, the SA market is expected to grow at a compounded annual growth rate of 8.2%, with an expected market reaching more than USD200 million by 2026[3]. At present, SA is mainly produced by a petrochemical route from maleic anhydride as the raw material. With increasing concerns about climate change, it is of great significance to develop green, sustainable and efficient technologies to produce SA from renewable biomass[4,5].

SA is an intermediate of the tricarboxylic acid (TCA) cycle, and it can be produced through both the oxidative and the reductive branch[6,7]. Many bacteria have the reductive TCA cycle and synthesize SA as an end product of anaerobic fermentation[8–11]. However, bacteria are generally not tolerant to acids and low pH, and this imposes a requirement for the addition of alkali for maintaining pH during the fermentation. This imposes a requirement for a large amount of

sulfuric acid in the extraction process[12]. There is therefore interest in establish a SA fermentation process in which the pH does not need to be controlled during the fermentation. Yeasts are highly tolerant to acid and are potential SA producers with high performance at low pH[13,14].

Several yeasts have been engineered for SA biosynthesis, including the ethanol-producing yeast *Saccharomyces cerevisiae*, *Issatchenkia orientalis*, and the strictly aerobic yeast *Yarrowia lipolytica*[13,15,16]. However, the yields of engineered yeasts are still relatively low. In these yeasts, the production of SA through the oxidative TCA pathway is accompanied by the release of $CO_2$ and the maximum theoretical yield is 0.65 g/g glucose. We previously constructed the oxidative SA-producing strains of *Y. lipolytica* by disrupting succinate dehydrogenase (SDH)[17–19]. Although the strains can produce SA with a high titer, the highest SA yield was only 0.53 g/g glycerol[20]. The reductive TCA branch fixes $CO_2$ for SA production with a higher theoretical yield of 1.12 g/g glucose. However, in the reductive TCA cycle, two moles of

[1]State Key Laboratory of Microbial Technology, Shandong University, 266237 Qingdao, P. R. China. [2]Marine Biology Institute, Shantou University, 515063 Shantou, P. R. China. [3]National Glycoengineering Research Center, Shandong University, 266237 Qingdao, P. R. China. [4]Department of Life Sciences, Chalmers University of Technology, Gothenburg SE41296, Sweden. [5]BioInnovation Institute, Copenhagen N DK2200, Denmark. ✉e-mail: houjin@sdu.edu.cn; qiqingsheng@sdu.edu.cn

NADH are needed for every mole of SA synthesized from pyruvate, but only one mole of NADH can be produced through the glycolytic pathway with the generation of one mole pyruvate. Thus, the reductive TCA pathway usually operates under anaerobic conditions as a NADH sink[21]. Due to the cofactor imbalance in the cytosol, the introduction of a reductive TCA branch did not boost the SA yield of yeasts. For example, the expression of fumarate reductases only resulted in 12.97 g/L SA with a yield of 0.13 g/g glucose in *S. cerevisiae*[22]. To increase the NADH availability, heterogenous transhydrogenase was expressed in reductive TCA expressed *I. orientalis* to generate NADH from NADPH. The strain produced SA with a titer of 89.0 g/L, and productivity 0.93 g/L/h[23]. Glycerol can produce more reducing equivalents of NADH than glucose during catabolism. Recently, Tran et al. engineered *I. orientalis* to use glycerol as a second carbon source for cytosolic NADH supply[24]. 109.5 g/L SA was produced and the SA yield was 0.65 g/g using both glucose and glycerol as substrates. Although there have been many efforts to increase the NADH supply, either the SA yields or titers of these engineered yeasts remain insufficient (Supplementary Table 1)[25–27].

Here, we show reconfiguration of the reductive TCA cycle in different subcellular compartments overcomes the shortage of cytosolic NADH supply and enables *Y. lipolytica* for high-efficient SA production at low pH using glucose as a carbon source (Supplementary Fig. 1). In pilot-scale fermentation, the engineered strain produces 111.9 g/L SA with a yield of 0.79 g/g glucose and a productivity of 1.79 g/L/h. Our study therefore paves the way for industrial production of biobased SA.

## Results

### SA biosynthesis by *Y. lipolytica* via the reductive TCA cycle

To construct the reductive TCA cycle (Fig. 1a), the key enzyme fumarate reductase (Frd) was introduced into *Y. lipolytica*. From literature reports[28,29] and homology analysis we identified six potential soluble Frds (Supplementary Table 2), and these were all expressed and evaluated in our previously constructed SDH-deficient strain PGC91 (*MatA, xpr2-322, axp-2, leu2-270, ura3-302, ΔSdh5::loxP, ΔAch1::loxP, YlPyc*)[20]. All tested Frds except ScOsm1 from *S. cerevisiae* improved the SA yield to different extents in glucose medium (Fig. 1b). Among these Frds, the Frd from *Trypanosoma brucei* (TbFrd) resulted in an SA yield of 0.62 g/g glucose, which was 29.2% higher than that of the control strain. We then cultivated this strain in a flask with a low rotation speed to set microaerobic conditions and found that although the SA titer was not high, the yield of SA increased to 0.72 g/g glucose, which exceeded the theoretical value of the oxidative TCA cycle and indicated that a reductive SA biosynthesis pathway was successfully established (Supplementary Fig. 2).

To enhance SA production in the PGC91-TbFrd strain, the metabolic pathway was optimized by the combined expression of genes involved in the reductive TCA cycle (Fig. 1c). The individual expression of phosphoenolpyruvate carboxykinase from *S. cerevisiae* (encoded by *ScPck*), fumarase from *E. coil* (encoded by *EcFum*), and endogenous malate dehydrogenase (encoded by *YlMdh1* and *YlMdh2*) increased SA production in PGC91-TbFrd. When *EcFum* and *YlMdh1* were co-expressed with *TbFrd* in the strain designated as PGC91-rT, the SA yield reached 0.85 g/g glucose, 30.8% higher than that of the strain with single TrFrd expression. However, the $OD_{600}$ of PGC91-rT was only 9.7 after 96 h cultivation, 52.5% lower than that of the starting strain PGC91 (Fig. 1d and e). Overall, these results demonstrated that the reductive SA biosynthesis pathway can be constructed in strictly aerobic *Y. lipolytica*, but its establishment affects cell metabolism and cell growth. Further overexpression of two key enzymes in the glyoxylate shunt, namely isocitrate lyase (encoded by *YlIcl1*) and malate synthase (encoded by *YlMls*) as well as endogenous glucose transport protein encoded by *YlYht1* and *YlYht4* did not improve cell growth and SA production.

### Adaptive laboratory evolution of engineered strain

Rational modification did not relieve the metabolic disorders. Therefore, *Y. lipolytica* PGC91-rT was subjected to adaptive laboratory evolution. Three independent evolutions were performed. The evolution process was maintained at a low rotation speed of 120 rpm in shaking-flasks to ensure microaerobic conditions and greater carbon flux into the reductive TCA cycle. During adaptive laboratory evolution, the glucose concentration in the YPD medium was increased gradually. After about 40 transfers (~120 generations), all three groups acquired slightly restored cell growth in the glucose medium (Fig. 2a and b). Most of the detected colonies showed increased SA production, but also produced more than 5 g/L malate as a by-product (Supplementary Fig. 3). Three evolved strains (one from each group) were selected and designated as PGC91-rTE1-1, PGC91-rTE2-1, and PGC91-rTE3-1. Strain PGC91-rT showed a growth defect on solid YPD medium, while the PGC91-rTE strains grew well (Fig. 2c). In shaking-flask cultures, the cell growth and SA production of these three strains were better than those of the un-evolved strain PGC91-rT (Fig. 2d and e).

### Genomic sequencing of the evolved strains revealed reduced flux to pentose phosphate pathway

The genome of the parent strain and two independent evolved clones from each group (including the previously selected clones) were sequenced to identify the mutations in the evolved strains. Notably, only 11 single nucleotide variations (SNVs) and small insertions/deletions (InDels) were identified in gene coding regions. Among all the mutations, a missense mutation in *Pgl1*, the gene encoding 6-phosphogluconolactonase (YALI0_C19085g), was found in all three groups (*Pgl1*[G75S] in group I and *Pgl1*[G74V] in group II and III) (Fig. 2f and Supplementary Data 1). Validation by PCR amplification and Sanger sequencing revealed no real mutations in any other genes aside from *Pgl1*.

Pgl1 is a key enzyme in the pentose phosphate pathway (PPP). The amino acids at positions 74 and 75 sites in Pgl1 were mutated to valine and serine from glycine. Although the amino acid sequence alignment showed that these two amino acid residues are not in the catalytic center, they are evolutionarily conserved among Pgl enzymes from different species (Supplementary Fig. 4). We used the online tool I-TASSER to simulate the protein structure of Pgl1, Pgl1[G75S], and Pgl1[G74V] (Fig. 3a). The mutation sites (blue labels) were located at the substrate binding pocket. We speculate the changes in the amino acid side chain from −H to $-CH_2-OH$ or $-CH-(CH_3)_2$ may affect the affinity of the enzyme for the substrate.

We initially tried to mutate the *Pgl1* gene of the PGC91-rT strain, but this was unsuccessful because of the poor growth of PGC91-rT and the low efficiency of homologous recombination. We then measured the enzyme activity of Pgl1[G75S] and Pgl1[G74V] in vitro (Supplementary Fig. 5). As shown in Fig. 3b, compared with the control Pgl1, the Pgl1[G75S] and Pgl1[G74V] mutants showed lower activity (29.8% and 35.9% lower, respectively). This proved that the *Pgl1* gene mutation in evolved strains caused a decrease in the catalytic activity of Pgl1, which in turn reduced the metabolic flux through the PPP. Consistently, the $NADPH/NADP^+$ ratios in strains PGC91-rTE1-1 (containing Pgl1[G75S]) and PGC91-rTE2-1 (containing Pgl1[G74V]) were 59.3% and 55.8% lower than that in strain PGC91-rT (Fig. 3c); and the $NADH/NAD^+$ ratios in PGC91-rTE1-1 and PGC91-rTE2-1 were 90.9% and 47.4% higher, respectively, than that in strain PGC91-rT. These results indicated that insufficient glycolysis (via the Embden−Meyerhof−Parnas pathway, EMP pathway) may be the cause of metabolic disorders in the strains overexpressing the reductive TCA cycle in the cytosol. When the activity of the PPP was reduced in the evolved strains, the greater carbon flux into the EMP pathway may have provided additional NADH and substrates for cell growth and SA synthesis (Fig. 3d).

To increase the supply of cytoplasmic NADH supply in the cytoplasm, *E. coli*-derived transhydrogenase and endogenous pyruvate decarboxylase bypass were introduced into the PGC91-rTE1-1 strain,

respectively[9,23,30]. However, transhydrogenase expression did not improve SA production, and the introduction of endogenous pyruvate decarboxylase bypass only increased SA titer slightly. The final SA titer of this strain was 23.4 g/L (Supplementary Fig. 6). It demonstrated further engineering cytoplasmic NADH supply cannot improve SA production significantly.

## Mitochondrial localization of reductive TCA facilitates SA biosynthesis

In mitochondria, NADH is generated through pyruvate dehydrogenation and the oxidative TCA pathway. In SDH-deficient *Y. lipolytica*, oxidative phosphorylation is partially disrupted, and the complete oxidative TCA pathway is not functional. Establishment of the reductive TCA cycle may release these inhibitions by re-directing metabolic flux into SA formation and also provide a sink for NADH. Therefore, we first attempted to localize Frd to the mitochondrial matrix (Fig. 4a).

The signal peptide of cytochrome c oxidase 5b in *S. cerevisiae* has been used frequently for mitochondrial localization of proteins[31]. The mitochondrial targeting sequence (MTS) of cytochrome c oxidase 5b in *Y. lipolytica* was predicted using the online tool MITOPROT[32]. Then MTS consisting of 23 amino acids was fused to the N-terminus of the green fluorescent protein sfGFP, and expressed in *Y. lipolytica* Po1f. The mitochondria were visualized by the mitochondria marker Mito-Tracker Red CMXRos. The fluorescence distribution of MTS-sfGFP was consistent with mitochondrial localization, while sfGFP without the signal peptide showed a cytosolic distribution (Fig. 4b). These results indicate that the N-terminal signal sequence of cytochrome c oxidase 5b was able to guide the target protein to the mitochondrial matrix of *Y. lipolytica*.

TbFrd[33], ScOsm1[29], and SfFcc3[34] are three different types of soluble Frds that are dependent on NADH, FADH₂, and quinol as electron donors, respectively. These enzymes show high structural similarity

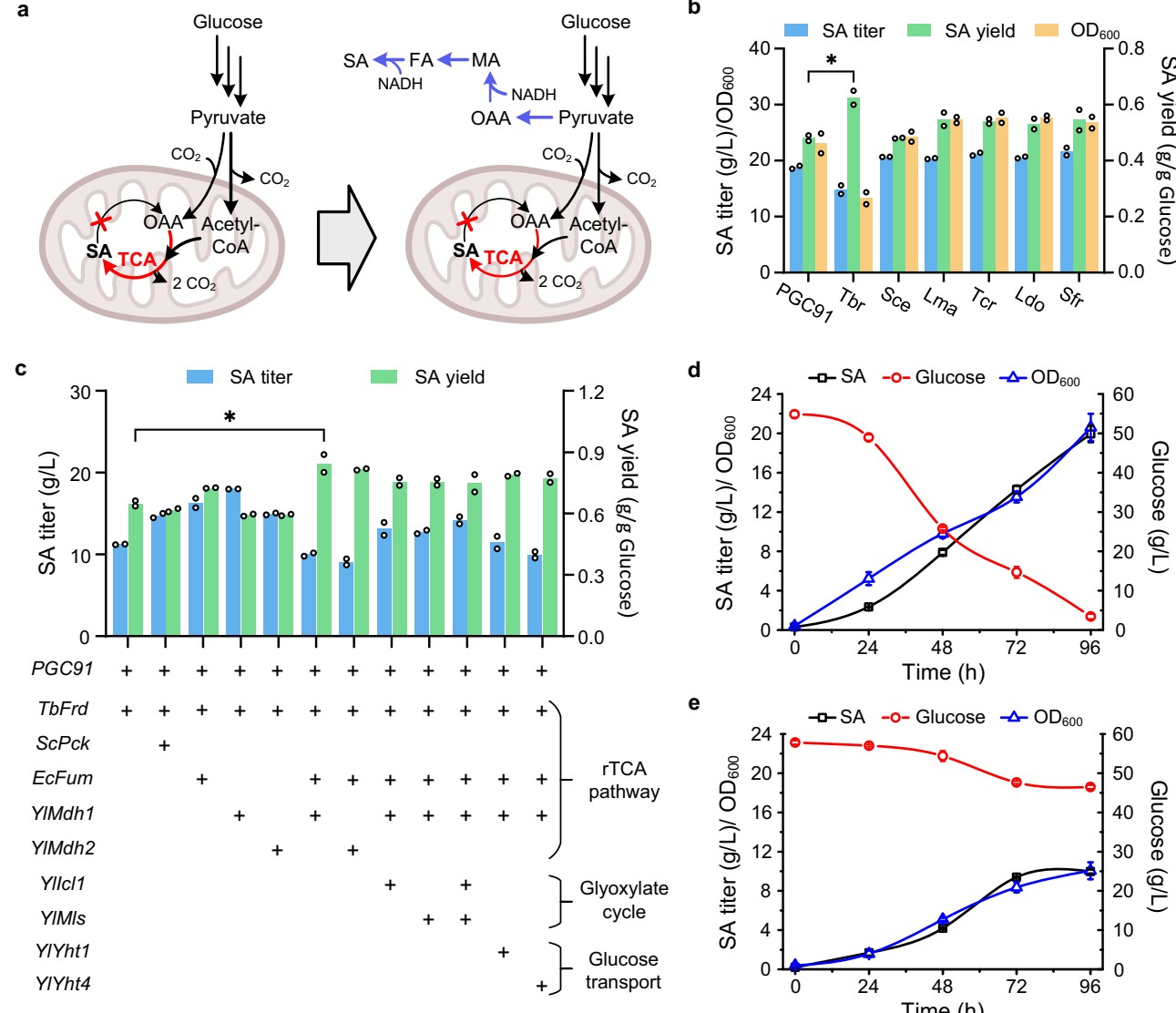

**Fig. 1 | Functional expression of the reductive TCA cycle in *Y. lipolytica*.**
**a** Construction of the reductive TCA cycle in SDH-deficient *Y. lipolytica* strain. Red arrows represent the oxidative TCA cycle, and blue arrows represent the reductive TCA cycle. Red X represents the deficiency of succinate dehydrogenase. OAA oxaloacetate, MA malate, FA fumarate, SA succinate, NADH nicotinamide adenine dinucleotide. **b** Evaluating different Frds on SA synthesis of PGC91 strain using glucose as the substrate at 220 rpm. **c** Combinatorial overexpression of genes

associated with the cytoplasmic reductive TCA cycle. Comparison of the physiological properties of the PGC91 (**d**) and PGC91-rT (**e**) strains during shaking-flask culture at 120 rpm. The initial concentration of glucose was 60 g/L. Data are presented as mean ± s.e.m. (*n* = 2 of **b** and **c**, 3 of **d** and **e** biologically independent samples). Statistical analysis was carried out by using Student's *t*-test (one-tailed; two-sample unequal variance; *$P < 0.05$, **$P < 0.01$, ***$P < 0.001$). Source data are provided as a Source Data file.

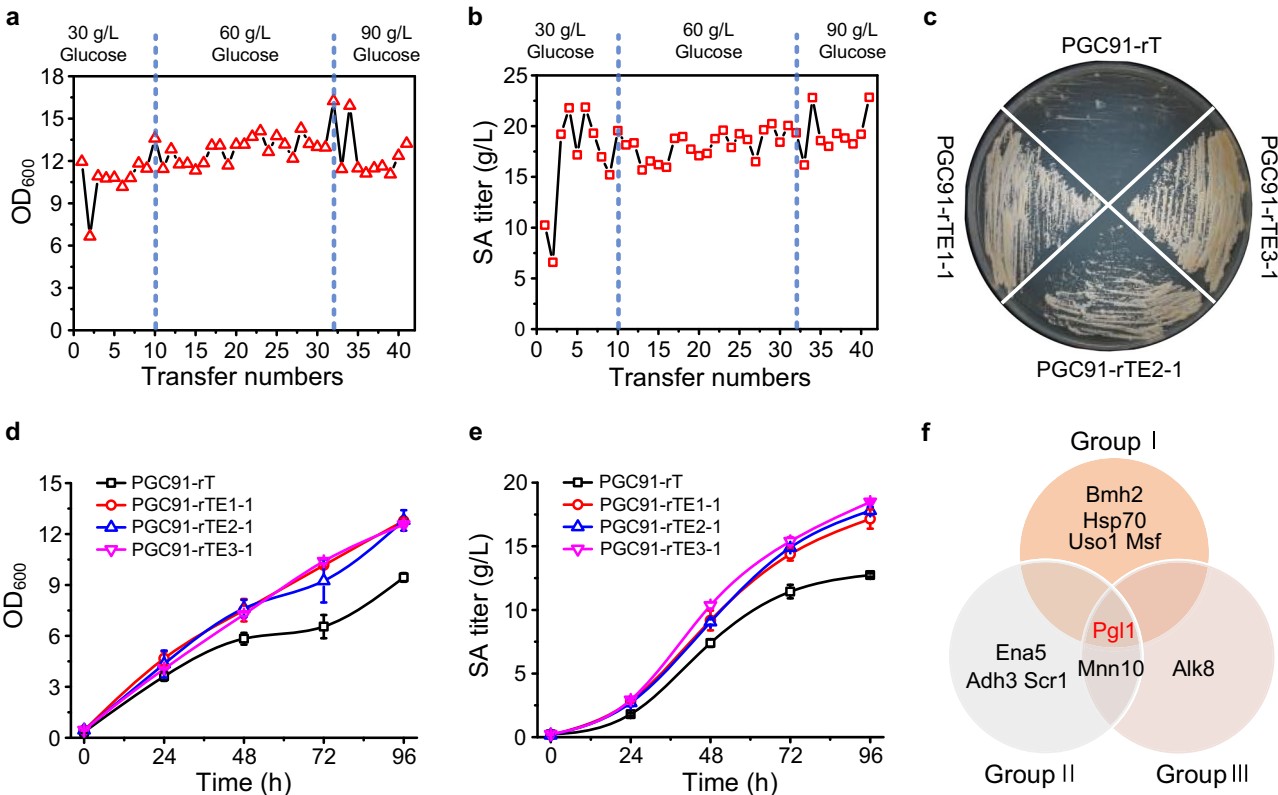

**Fig. 2 | Adaptive laboratory evolution.** Monitoring of $OD_{600}$ (**a**) and SA (**b**) titer during adaptive evolution of PGC91-rT in glucose medium. **c** Growth recovery of PGC91-rTE strains on YPD plates. Comparison of cell growth (**d**) and SA production (**e**) between the starting strain PGC91-rT and the evolved PGC91-rTE strains in YPD medium at 120 rpm. The initial concentration of glucose was 40 g/L. Data are presented as mean ± s.e.m. (three biologically independent samples were adopted). **f** Mutations (SNVs and InDels) detected via genome sequencing from three groups. For more details see Supplementary Data 1. Source data are provided as a Source Data file.

(Supplementary Fig. 7). These Frds were selected from the above-mentioned six candidates (Supplementary Table 2), and then fused with MTS and overexpressed in the PGC91-rTE1-1 strain. As shown in Fig. 4c, mitochondrial localization of TbFrd significantly improved the SA synthesis ability of PGC91-rTE1-1, and its titer and yield of SA reached 30.8 g/L and 0.83 g/g glucose, respectively. In contrast, the expression of another copy of cytosolic TbFrd in the PGC91-rTE1-1 strain caused a decrease in SA titer and $OD_{600}$. All PGC91-rTE1-1 derivatives harboring additional TbFrd overexpression no longer accumulated by-product malate. Next, the Frd activity in purified mitochondria from PGC91-rTE1-1 and PGC91-rTE-mTbFrd cells was analyzed. Only the mTbFrd-overexpressing strain was able to produce SA using fumarate and NADH as the substrates, confirming the functional expression of mTbFrd (Supplementary Fig. 8). The above data show that the mitochondrial localization of NADH-dependent Frd was highly advantageous for improving SA production in *Y. lipolytica*.

Other enzymes involved in the reductive TCA cycle were also overexpressed in the mitochondria of PGC91-rTE-mTbFrd (Fig. 5a). The combination of YlMdh2 and mTbFrd led to the high SA production at 41.4 g/L, which was 35.7% higher than that of the control strain (Fig. 5b). Subsequently, we generated strain Hi-SA0 overexpressing the SA transporter gene *SpMae1*. The SA titer and yield of Hi-SA0 reached 46.7 g/L and 0.86 g/g glucose, respectively. To further improve SA production, we overexpressed the genes involved in the oxidative TCA cycle (Fig. 5a). When YlScs2 was overexpressed in the Hi-SA0 background (yielding strain Hi-SA1), the SA titer and yield were 51.7 g/L SA and 0.87 g/g glucose, respectively (Fig. 5b). This suggested that enhanced metabolic flux through the oxidative TCA cycle benefits reductive SA biosynthesis. When the enzymes YlPyc and YlFum were localized to the mitochondrial matrix of Hi-SA0 (in strain Hi-SA2), the

SA titer and yield in shaking-flask culture were 74.4 g/L and 0.94 g/g glucose, respectively; these values were 295.7% and 42.4% higher than those in PGC91-rTE1-1 (Fig. 5b). Targeting the glyoxylate cycle YlMls and YlIcl2 to the mitochondria did not further improve SA production. Hi-SA1 and Hi-SA2 were able to grow and accumulate SA continuously in long-term shaking-flask fermentation (Supplementary Fig. 9). After extended culture for 192 h, Hi-SA2 produced SA at 103.2 g/L, with a yield of 0.92 g/g glucose (Supplementary Fig. 9c).

[13]C-Metabolic flux analysis was then performed to analyze the metabolic flux distribution in Hi-SA2 (Supplementary Fig. 10) at different rotational speeds (220 and 120 rpm). Carbon flux through the oxidative PPP was similar for different dissolved oxygen, which was calculated to be about 33, lower than the value previously reported for wild-type *Y. lipolytica*[35]. This result was supported by the low activity of Pgl1. The flux from glyceraldehyde 3-phosphate to 3-phosphoglycerate was 182. It was reported that ≥84% of the SA produced by an SDH-deficient *Y. lipolytica* strain was formed through the oxidative TCA cycle[27]. As shown in Supplementary Fig. 10, the ratio of metabolic flux between reductive and oxidative TCA cycles in *Y. lipolytica* Hi-SA2 strain is affected by dissolved oxygen level. At a high speed of 220 rpm, only about 30% of the carbon flux into SA biosynthesis was derived from the reductive TCA cycle, while more than half flux into SA biosynthesis was derived from the reductive TCA cycle when the rotational speed was 120 rpm.

**Scaled-up SA production by *Y. lipolytica* Hi-SA2**
Scaled fermentation of Hi-SA2 in YPD medium was carried out in a 5-L bioreactor. Through optimization, the fermentation conditions were set as follows: airflow rate of 1 vvm, stirring speed of 400 rpm, and no pH control. The biomass and SA titer reached 33.1 and 80.9 g/L,

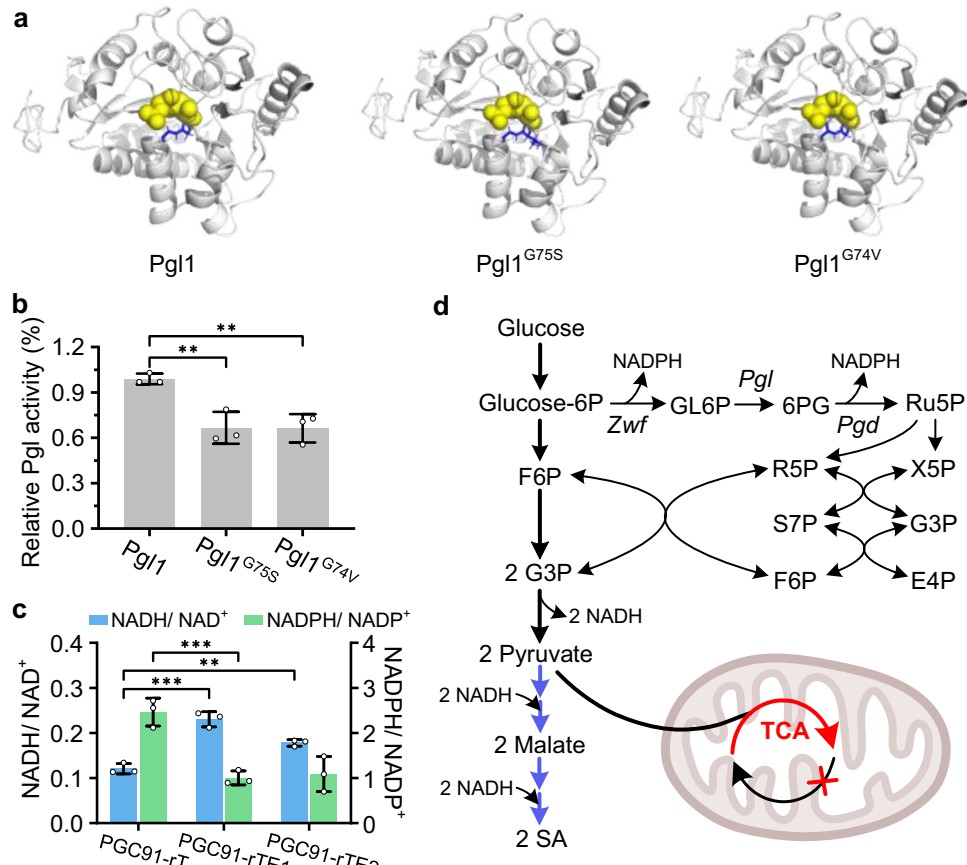

**Fig. 3 | 6-phospho-gluconolactonase mutation redistributes the metabolic flux in PGC91-rTE strains. a** Protein structural analysis of Pgl1 mutants. **b** In vitro enzyme activity assay of Pgl1, Pgl1[G75S], and Pgl1[G74V]. **c** In vivo NADH/NAD+ and NADPH/NADP+ ratios of PGC91-rT and PGC91-rTE strains. **d** Schematic position of the Pgl1 mutation in PPP and SA biosynthesis pathway. *Zwf* encoding glucose 6-phosphate dehydrogenase, *Pgl* encoding 6-phospho-gluconolactonase, *Pgd* encoding gluconate 6-phospho dehydrogenase. F6P fructose 6-phosphate, X5P xylulose 5-phosphate, S7P sedoheptulose 7-phosphate, GL6P gluconolactone 6-phospho, 6PG gluconate 6-phospho, Ru5P ribulose 5-phosphate, R5P ribose 5-phosphate, G3P glyceraldehyde 3-phosphate, and X5P xylulose 5-phosphate. NADH nicotinamide adenine dinucleotide, NADPH nicotinamide adenine dinucleotide phosphate. Data are presented as mean ± s.e.m. (*n* = 3 biologically independent samples). Statistical analysis was carried out by using Student's *t*-test (one-tailed; two-sample unequal variance; *P < 0.05, **P < 0.01, ***P < 0.001). Source data are provided as a Source Data file.

respectively, and the final pH decreased to 3.31 (Supplementary Fig. 11). The SA yield was about 0.90 g/g glucose. Then, to meet the requirement of industrialization, a modified minimal medium CM1 supplemented with corn steep powder as a cheap nitrogen source was used instead of the expensive YPD medium (Supplementary Fig. 12). When strain Hi-SA2 was cultivated in the 5-L bioreactor in modified CM1 medium, the SA titer and yield were 80.2 g/L and 0.69 g/g glucose, respectively (Supplementary Fig. 13).

To explore the industrial application potential of Hi-SA2, SA fermentation was scaled up to a 50-L bioreactor with the modified CM1 medium. The fermentation conditions were 30 °C, 300 rpm, 0.5 vvm, and no pH control. At the initial stage of fermentation, the biomass increased slowly, and the SA accumulation rate was also low, but then SA production increased rapidly (Fig. 6a). The cells reached the stable growth phase after 18 h of culture, and they continued to produce SA. At 62 h, the SA production reached a maximum SA titer of 111.9 g/L, and the pH of the fermentation medium dropped to 2.49. Glucose was fully consumed at the end of the fermentation period, and almost no by-products were detected. The overall SA productivity and yield were 1.79 g/L/h and 0.79 g/g glucose, respectively. The bio-based SA from the acidic broth of the fed-batch SA fermentation was extracted and purified (Fig. 6b). This process did not require an acidification step using sulfuric acid so simply through a process of filtration, distillation, and crystallization, SA crystals with a purity of 93.5% could be obtained.

## Discussion

Biosynthesis of SA through the reductive TCA cycle was thought to be the best choice because of its higher theoretical yield[36]. However, the reductive SA synthetic pathway generally produces SA in anaerobic or oxygen-limited conditions. Although this pathway has been constructed in facultative anaerobic yeast, such as *S. cerevisiae* and *I. orientalis*[14,22,37], either the titer or yield was relatively low. Here we constructed a functional reductive TCA cycle in the strictly aerobic yeast *Y. lipolytica*. Relocation of the reductive TCA cycle to the mitochondria enabled the engineered yeast strain to utilize NADH generated in the oxidative TCA pathway for SA biosynthesis. The final engineered strain produced SA with a yield of 0.90 g/g glucose in a bench-top bioreactor. When scaled up in a pilot-scale fermentation, the SA titer and yield were able to reach 111.9 g/L and 0.79 g/g glucose, respectively. Compared with other SA producers at low pH, this engineered strain produced SA with a higher yield, titer, and productivity (Supplementary Table 1).

Facultative anaerobic microorganisms tend to accumulate reductive compounds, such as glycerol and ethanol under hypoxic conditions[38]. During optimization of *I. orientalis*, the first constructed strain accumulated up to 9.5 g/L ethanol besides SA, which necessitated the removal of the pathway generating this reductive compound[24]. In contrast, *Y. lipolytica* is a strictly aerobic yeast, so the natural fermentative pathway is not active. The engineered strains of

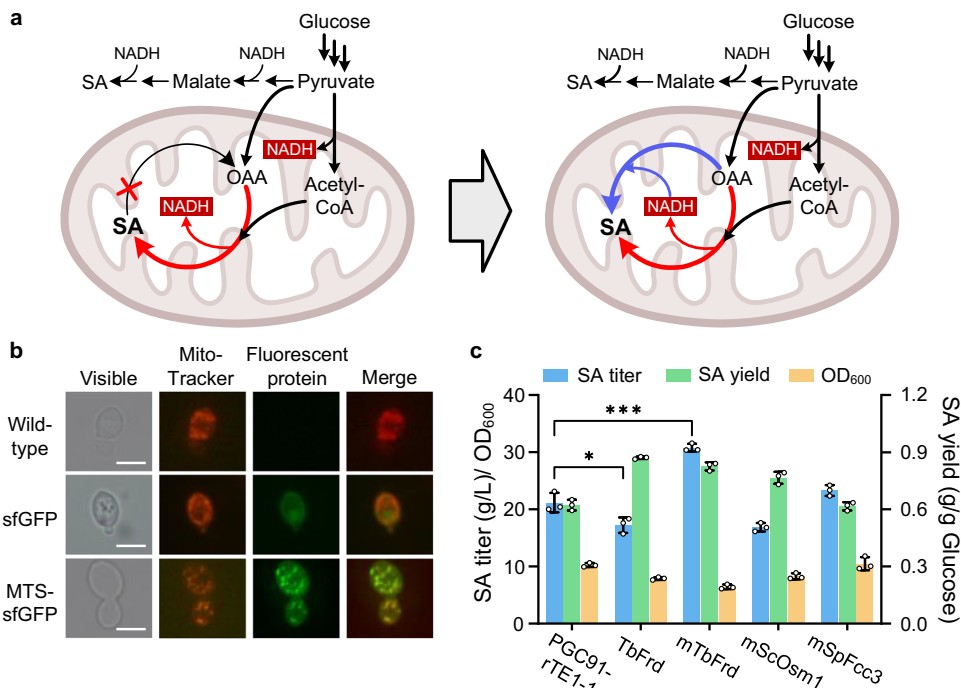

**Fig. 4 | Subcellular localization of fumarate reductase. a** Schematic representation of the mitochondrial localization of the reductive TCA cycle. Red arrows represent the oxidative TCA cycle, and blue arrows represent the reductive TCA cycle. **b** Functional validation of the mitochondrial targeting sequence from cytochrome c oxidase 5b. Fluorescence microscopy of yeast strains expressing sfGFP and strained with Mito-Tracker Red CMXRos. Green fluorescent protein sfGFP with or without mitochondrial targeting sequence were expressed in wild-type strain

Po1f, respectively. $n = 3$ biologically independent samples. Scale bar: 2 μm. **c** Expression of cytosolic and mitochondrial soluble Frd in the evolved strain PGC91-rTE1-1 to enhance SA production. The initial concentration of glucose was 60 g/L. Data are presented as mean ± s.e.m. ($n = 3$ biologically independent samples). Statistical analysis was carried out by using Student's $t$-test (one-tailed; two-sample unequal variance; *$P < 0.05$, **$P < 0.01$, ***$P < 0.001$). Source data are provided as a Source Data file.

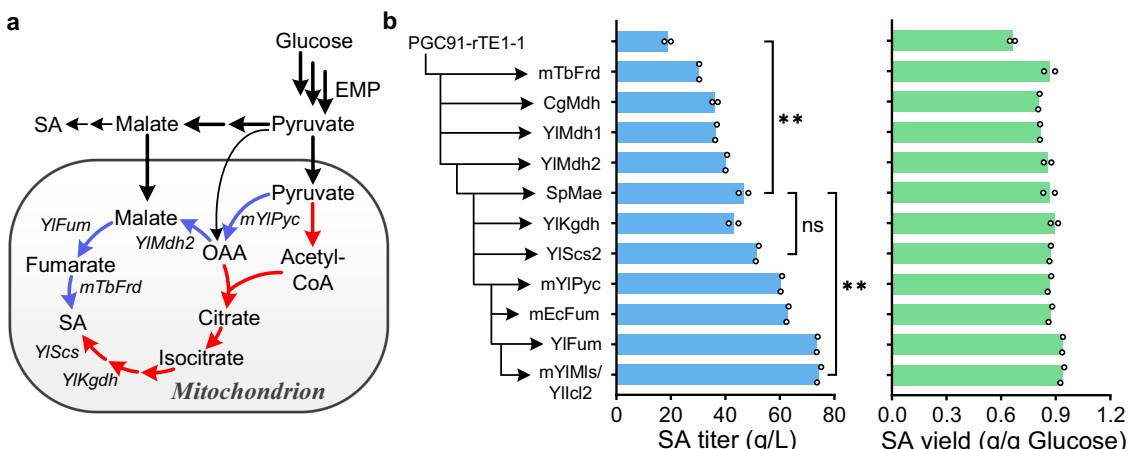

**Fig. 5 | Optimization of reductive SA biosynthetic pathway in mitochondrial matrix of *Y. lipolytica*. a** Metabolic reprogramming of mitochondria as an intracellular workshop for SA biosynthesis. Red arrows represent the oxidative TCA cycle, and blue arrows represent the reductive TCA cycle. *mYlPyc* encoding mitochondrial pyruvate carboxylase; *YlMdh2* encoding malate dehydrogenase; *YlFum* encoding fumarase; *mTbFrd* encoding mitochondrial fumarate reductase; *YlScs* encoding succinyl-CoA synthetase; *YlKgdh* encoding α-ketoglutarate

dehydrogenase. EMP Embden–Meyerhof–Parnas pathway. **b** SA titers and yields of mitochondrial localization of the enzymes in the reductive and oxidative TCA cycle in PGC91-rTE-mTbFrd strain. The initial concentration of glucose was 80 g/L. Data are presented as mean ± s.e.m. ($n = 2$ biologically independent samples). Statistical analysis was carried out by using Student's $t$-test (one-tailed; two-sample unequal variance; *$P < 0.05$, **$P < 0.01$, ***$P < 0.001$; ns represents no significant difference). Source data are provided as a Source Data file.

*Y. lipolytica* produced SA as the sole reductive compound, which also simplified the downstream SA purification process[12].

The starting strain in this study was an SDH-deficient strain of *Y. lipolytica* that can accumulate SA with a yield of 0.41 g/g glycerol[20]. Artificial construction of the reductive SA production pathway resulted in the strain *Y. lipolytica* PGC91-rT, with an SA yield increased to

around 0.85 g/g glucose. However, this strain showed a growth-deficient phenotype, even in the YPD medium. This phenomenon has not been found in facultative anaerobic yeasts, probably because of their natural fermentative pathways. Laboratory evolution of PGC91-rT restored cell growth but did not markedly improve the SA titer and yield. Genomic analyses revealed mutations in the key gene *Pgl1*,

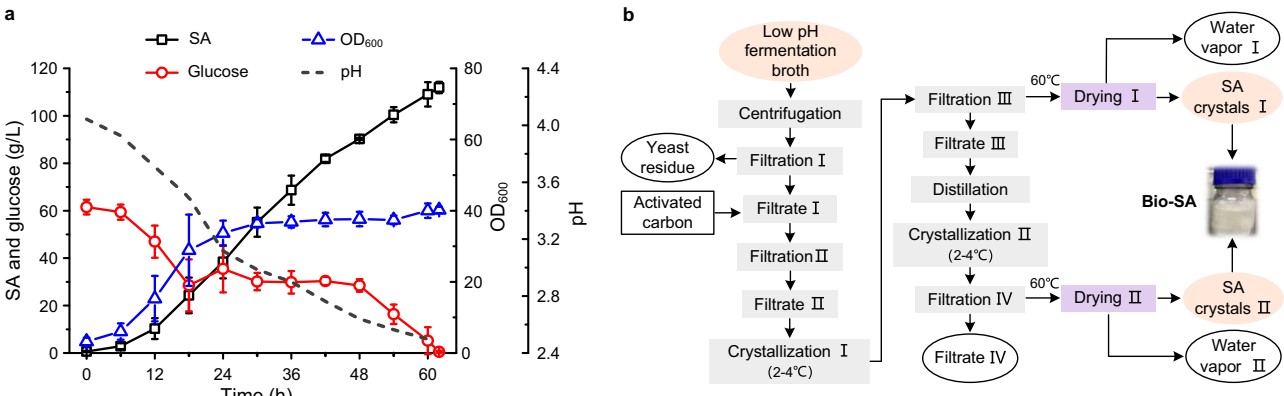

**Fig. 6 | Scale-up SA production of engineered *Y. lipolytica* strain Hi-SA2 at low pH. a** Fermentation profile of Hi-SA2 strain in 50-L bioreactor. **b** Schematic diagram of the modified SA recovery process. Data are presented as mean ± s.e.m.

($n$ = 3 biologically independent samples). Source data are provided as a Source Data file.

encoding a 6-phospho-gluconolactonase that generates the substrate for the PPP. Wild-type *Y. lipolytica* has a high flux through the oxidative PPP to generate the large quantities of NADPH required for lipogenesis[39,40]. Pgl1 mutations may redirect the metabolic flux from the PPP to glycolysis and generate more NADH than NADPH. Because strain PGC91-rT did not grow in a minimal medium, we only measured the metabolic fluxes of the final strain. [13]C-MFA results indicated that the flux to the PPP in microaerobic conditions was reduced to 33 (Supplementary Fig. 10), much lower than in wild-type strain[35]. However, the NADH supply was still a bottleneck in the PGC91-rTE strains. Further engineering of NADH regeneration, including expression of transhydrogenase and bypassing pyruvate in the cytosolic, did not result in an obvious increase in SA production.

Accumulation of SA via the reductive TCA cycle consumes more NADH than does its generation from glucose. The SA titer of PGC91-rT was below 10 g/L because of the shortage of NADH in the cytosol (Fig. 1e). In the mitochondria, three molecules of NADH can be produced through one cycle of the oxidative TCA cycle, but the mitochondrial membrane restricts the transport of NADH from mitochondria into the cytosol. We therefore localized the reductive TCA cycle to the mitochondria such that NADH generated from the oxidative TCA cycle could be used for SA production via the reductive TCA cycle. In *Y. lipolytica* PGC91-rT, *YlSdh5* deletion eliminated the complete oxidative TCA cycle because complex II of oxidative phosphorylation was disrupted and oxidation of NADH was inhibited. Localization of the reductive TCA cycle in mitochondria may release the inhibition of SDH deficiency by re-using the NADH generated from the oxidative TCA cycle for SA synthesis. Our experimental results show that this strategy is efficient. Simply expressing TbFrd in the mitochondria increased SA production from 20.2 to 30.4 g/L (Fig. 4c), and combined overexpression of the genes involved in reductive TCA cycle in the mitochondria increased the SA production to 74.4 g/L with a yield of 0.94 g/g in batch fermentation (Fig. 5c). These results indicate that, during metabolic engineering, attention should be paid not only to the cofactor balance and regeneration at the whole-cell level but also at the subcellular level.

We compared the SA /$OD_{600}$ levels of final engineered strain Hi-SA2 in a flask, 5-L, and 50-L bioreactors (7.82, 2.45, and 2.78, respectively), it was indicated that the SA production capacity per unit biomass is decreasing in the scale-up fermentation. This result also shows the potential for further improvement opportunities via fermentation optimization and scale-up in the future.

In summary, we demonstrate that localization of the reductive TCA cycle in the mitochondria balances NADH generation and consumption, thereby redirecting metabolic flux to SA biosynthesis in

*Y. lipolytica*. This results in an engineered strain with high SA titer and yield. The findings of this study have paved the way for the industrialization of biobased SA production and may provide a reference for the synthesis of other reducing products in aerobic yeast.

## Methods

### Strains, reagents, and medium

*Y. lipolytica* strain PGC91 (*MatA, xpr2-322, axp-2, leu2-270, ura3-302, ΔSdh5::loxP, ΔAch1::loxP, YlPyc*) derived from Po1f (*MatA, xpr2-322, axp-2, leu2-270, ura3-302*) was used as the background strain for all genetic manipulations and strain construction[20]. The yeast strains used in this study are listed in Supplementary Table 2. All restriction enzymes were purchased from Thermo Fisher Scientific (Shanghai, China). PCR amplifications were performed using PrimeSTAR Max DNA Polymerase (TaKaRa, Beijing, China) or GoldenStar T6 DNA polymerase (TsingKe, Beijing, China). DNA gel purification and plasmid extraction kits were purchased from Omega Bio-tek. All chemical standards were purchased from Sigma-Aldrich unless stated otherwise.

Luria-Bertani (LB) medium (10 g/L tryptone, 5 g/L yeast extract, 10 g/L NaCl) was used for routine cultivation of *E. coli*, and YPDG medium (10 g/L yeast extract, 20 g/L tryptone, 20 g/L glycerol, 20 g/L glucose) was used for cultivation of *Y. lipolytica*. When required, 50 μg/mL ampicillin, 400 μg/mL hygromycin B, and 350 μg/mL nourseothricin were added. A yeast synthetic defined premixes-SD base (TAKARA, Beijing, China) and suitable amino acid dropout mixes were used for the screening of auxotrophic transformants of *Y. lipolytica*. For fermentation of SA, the modified YPD medium (10 g/L yeast extract, 20 g/L tryptone, 40−80 g/L glucose), DM1 medium (0.71 g/L $KH_2PO_4$, 6 g/L $Na_2HPO_4 \cdot 12H_2O$, 11.77 g/L $NH_4Cl$, 0.65 g/L $(NH_4)_2SO_4$, 0.32 g/L $MgSO_4$, 1.8 g/L citrate, 0.11 g/L $CaCl_2$, 6.0 g/L corn steep powder/ yeast extract, 40−80 g/L glucose) or CM1 medium (0.8 g/L $Na_2HPO_4 \cdot 12H_2O$, 3.6 g/L $KH_2PO_4$, 1.2 g/L $MgSO_4 \cdot 7H_2O$, 2.8 g/L $(NH_4)_2SO_4$, 6.0 g/L corn steep powder/ yeast extract, 40−80 g/L glucose) were used.

### Plasmid construction and genetic engineering

*E. coli* DH5α was used for plasmid construction. Codon-optimized genes were synthesized by General Bio-tek (Chuzhou, China). Other native promoters, genes, and terminators were cloned from the genomic DNA of *Y. lipolytica* Po1f. The detailed information on genes/enzymes used in this study is listed in Supplementary Table 3. The plasmids, oligonucleotides, and genes used in this study are listed in Supplementary Data 2–4, respectively.

Superfolder GFP (sfGFP) encoding gene was amplified and assembled with *Bsp119*I digested YLEP-leu to obtain YLEP-leu-sfGFP.

Using the genomic DNA of Po1f and sfGFP genes as templates, the MTS sequence and sfGFP were amplified. The assembled fragment of MTS-sfGFP was then obtained by overlap extension PCR. MTS-sfGFP was assembled with *Bsp119*I digested YLEP-leu to obtain YLEP-leu-MTS-sfGFP.

Target genes were amplified from the corresponding plasmids or genomic DNA, and then assembled with restriction enzymes digested expression vectors including pKi-1, 113-GPD-TEF, pKi-hyg, and JMP-nat-GPD-TEF[41–43]. The resulting plasmids could be PCR assembled or digested to obtain integration fragments containing expression cassette and selection marker.

A homology-independent genome integration approach was used for the gene overexpression in *Y. lipolytica*[43]. Linearized DNA fragments carrying the target gene expression cassette were introduced into *Y. lipolytica* cells by lithium acetate transformation, and randomly integrated into the genome by endogenous NHEJ repair. Positive transformants were screened on appropriate solid plates and verified by colony PCR. Ten transformants were picked to select engineered strains with higher SA titer and yield by shaking flasks fermentation. Then the fermentation of high-SA producing strains was repeated.

To recirculate the selection marker genes, plasmids pUB4-Cre or YLEP-nat-Cre were introduced into the *Y. lipolytica*-engineered strains. Cre recombinase was constitutively expressed in a YPDG medium with corresponding antibiotics at 30 °C for 1–2 days. The positive recombinants were obtained through screening plate and colony PCR. Final strains were cultivated in YPDG for 2–3 days to remove the Cre-expressed plasmid.

## Screening and structural analysis of fumarate reductases
The amino acid sequences of fumarate reductases from different species were obtained from GenBank. Sequence homology was analyzed online using BLASTp. Protein homology modeling was performed by the online tool SWISS-MODEL (https://swissmodel.expasy.org/), and PyMOL was used for visualization.

## Adaptive laboratory evolution
A single colony of PGC91-rT strain was activated overnight and inoculated into YPD (10 g/L yeast extract, 20 g/L peptone, 30–100 g/L glucose) medium with three replicates. The conditions of evolution were carried out at 120 rpm and 30 °C in shaking-flask. Initial substrate concentration was gradually increased stepwise from 30, 60, to 90 g/L. When the growth reached the logarithmic phase ($OD_{600}$ reached about 5.0), the cells were transferred to a fresh YPD medium with an initial $OD_{600}$ of 0.5. The cultivation times needed for each round were about 36–48 h. During evolution, culture samples were taken every 12 h to detect $OD_{600}$ and SA production. About 40 times transfers were obtained. After the cell growth and SA accumulation were stable, the single clones were isolated and their SA production performance was detected. The three evolved independent culture lines were named PGC91-rTE1, PGC91-rTE2, and PGC91-rTE3.

## Genome sequencing analysis
Seven strains were selected to perform genome re-sequencing, including the control strain PGC91-rT and the evolved strains group I (PGC91-rTE1-1 and 1-2), group II (PGC91-rTE2-1 and 2-2) and group III (PGC91-rTE3-1 and 3-2). Genomic DNA was extracted and broken into 200-300 bp fragments using a Bioruptor ultrasonic fragmentation machine. After repairing the sticky end, base "A" was added to the 3′ end. The DNA linkers containing the Index sequence were added to both ends of the DNA fragments by TA ligation. The libraries were quantified by Qubit 2.0 Fluorometer (Thermo Scientific). The whole genome was sequenced using Illumina HiSeq/Nova 2x150bp at Azenta Life Sciences (Suzhou, China). The re-sequencing data were processed to obtain the original data, filtered to remove connectors, decontaminated, and compared with the reference genome. By comparing the

results, the repetitive sequences due to PCR amplification in each library were removed, and the single nucleotide site variants (SNVs), and insertion/deletion (InDels) relative to the reference genome were calculated. Then the missense mutations in coding regions of evolutionary strains were obtained by comparing them with the control strain. These results are shown in Supplementary Data 1.

## Heterologous expression and activity assay of Pgl1
The *Pgl1* gene (YALI0C19085p) was amplified from genomic DNA of PGC91-rT, the Pgl1 mutant genes *Pgl1^G75S* and *Pgl1^G74V* were obtained from genomic DNA of PGC91-rTE1-1 and PGC91-rTE2-1, respectively. These genes were cloned into the expression vector of pET-28a and transformed into *E. coli* BL21. After cell disruption, SDS–PAGE analysis and the purification of target proteins were performed.

To determine the activity of Pgl1[44], a reaction system containing 50 μM glucose 6-phosphate, 0.2 mM NADP, 25 mM HEPES (pH 7.1), 2 mM $MgCl_2$ and 1.75 U yeast glucose 6-phosphate dehydrogenase was incubated at 30 °C. When A340 reached a plateau, 0.5 U/mL of gluconate 6-phosphate dehydrogenase and the Pgl1 proteins (0.5–5 mU) were added, and A340 was further measured for ~10 min. For the determination of spontaneous hydrolysis of gluconolactone 6-phosphate, the buffer in the same volume as the probe without protein was added. One unit of enzyme is the amount that hydrolyzes 1 μmol of gluconolactone 6-phosphate per min under these conditions. The protein concentration was measured by Nanodrop to normalize the results.

## Identification of MTS and mitochondrial localization of enzymes
An MTS consisting of 23 amino acids (MFALRRSLLSAGRIARPQQVARF) from cytochrome c oxidase 5b was identified by using the online tool MITOPROT (https://ihg.gsf.de/ihg/mitoprot.html). To verify the function of the MTS, YLEP-sfGFP, and YLEP-MTS-sfGFP plasmids were transformed into *Y. lipolytica* Po1f, respectively. Yeast cells expressing sfGFP and MTS-sfGFP were cultivated in SD-medium lacking leucine for 36 h. Before imaging, yeast cells were stained with 1 μM Mito-Tracker Red CMXRos (Beyotime, Shanghai, China) for 30 min and resuspended in SD-medium. The cells were dropped onto microscope slides and then viewed with an Eclipse 80i microscope (Nikon).

For mitochondria localization of the enzymes, subcellular localization of the target enzymes was first analyzed by online tools MITO-PROT and Softberry (http://www.softberry.com/berry.phtml). For cytoplasmic proteins, MTS was added to their N-terminus of the proteins to achieve mitochondrial relocation. For the enzymes which itself has MTS, the original sequences of the enzymes were expressed.

## Batch fermentation in shaking flasks
SA-produced strains were pre-cultured in YPDG for 24–30 h and then transferred into 250 mL shaking flasks with 50 mL YPD medium. The fermentation of SA was carried out at 30 °C and 120–220 rpm for 72–192 h. Initial concentration of glucose was 40–80 g/L, and 1–2 mL glucose stock (500 g/L) was once fed when needed. All the fermentation conducted in shaking flasks did not add any alkali to maintain a neutral pH. Samples were taken every 24 h to measure biomass, residual glucose, and organic acids.

## Fed-batch fermentation in bioreactors
Hi-SA2 strain from −80 °C was streaked and cultivated on a YPDG solid plate for 36 h. The single colonies were inoculated into 50 mL shaking flasks with 10 mL YPDG medium at 30 °C and 220 rpm. After 24 h cultivation, culture (2%) was inoculated into 200 mL YPD medium in 1 L shaking flasks as seed culture at 30 °C and 220 rpm. Second seed culture (5%) was then inoculated into a 5-L bioreactor (BXBIO, Shanghai, China) to start SA fermentation. The optimal conditions for fed-batch fermentation were 30 °C, 400 rpm, and 1 vvm. Natural pH was maintained without adding any addition of alkali. The modified YPD

and CM1 medium with 60–80 g/L glucose were used as the initial batch medium. The glucose concentration was measured every 6 h, and when the glucose concentration fell below 10 g/L, the glucose stock (500 g/L) was supplemented. At the same time, the SA and biomass of the fermentation broth were detected.

For scaled-up fermentation in 50-L bioreactor (BXBIO, Shanghai, China), a single colony of *Y. lipolytica* strain Hi-SA2 was inoculated into 50 mL YPDG medium, and cultured for 24 h. Then, the cells were subcultured into a modified CM1 medium containing 40 g/L glucose and grown for 24 h. Seed culture (5%) was then added into the bioreactor containing 34 L of CM1 medium with 60 g/L glucose and 6 g/L corn steep powder. The cells were cultivated at 30 °C with 300 rpm, 0.5vvm. Antifoam was added if necessary. Natural pH was maintained without adding any addition of alkali. The pH changes were monitored by on-line pH analyzer of fermenter. The glucose stock (500 g/L) was supplemented to maintain glucose concentration of 30 g/L. At the later stage of fermentation, there was no feeding, so that the glucose was completely consumed.

### ¹³C-MFA

To investigate the flux distribution of SA-overproducing strain Hi-SA2, we performed $^{13}C$- metabolic flux analysis ($^{13}C$-MFA)[45]. Minimal CM1 medium with 10 g/L 100% 1-$^{13}C_1$ glucose as feeding substrate was used for cell cultivation. Cells at the exponential growth phase were harvested and washed twice with the synthetic medium. They were then hydrolyzed in 6 M HCl for 24 h at 120°C. The resulting proteinogenic acids were incubated with *N*-(tertbutyldimethylsilyl)-*N*-methyl-triuoroacetamide containing tert-butyldimethylchlorosilane in acetonitrile at 105 °C for 1 h, which was then analyzed by GC–MS (Agilent Technologies, Santa Clara, USA) equipped with a DB-1 column (Agilent Technologies). Data obtained from GC-MS were corrected by reducing the natural abundance ratios of C, H, O, N, and Si isotopes. The metabolic model was simulated and analyzed using the software package $^{13}C$-Flux[46].

### Metabolite extraction and quantification

Metabolites such as glucose, glycerol, and organic acids in the fermentation broth were determined by high-pressure liquid chromatography (HPLC) equipped with an Aminex HPX-87H column (Bio-Rad, Inc., Hercules, CA) and a refractive index detector. Before analysis, the extract solution was filtered through a 0.22 μm filter. The analysis was performed using 5 mM $H_2SO_4$ as mobile phase at 0.6 mL/min, and the column temperature was 65 °C.

Cellular NADH/NAD⁺ and NADPH/NADP⁺ were quantified using the Boxbio Coenzyme I NAD (H) Content Assay Kit (Catalog no. AKCO001C) and Coenzyme II NADP (H) Content Assay Kit (Catalog no. AKCO028C), respectively. Yeast cells (the number is about $10^4$) from 30 h cultures were centrifuged and resuspended in PBS buffer. Samples were individually extracted with the extraction buffers and all subsequent steps were performed according to the manufacturer's instructions.

To detect the activity of fumarate reductase in vitro, mitochondria were isolated from PGC91-rTE and PGC91-rTE-mTbFrd[6]. The yeast cells were resuspended in 0.1 M EDTA, pH 8.0, 0.1 M β-mercaptoethanol, and incubated at 30 °C for 30 min. After washing, cells were treated with Lyticase (Solarbio, Beijing, China) for 1 h in 25 mL of enzymatic hydrolysis buffer (0.1 M phosphate buffer, pH 7.4, 0.25 M MgCl₂, 0.9 M sorbitol) at 30 °C. Ice-cold glass beads were added, and the mixture was shaken for 5 min. Spheroplasts were homogenized in lysis buffer (200 mM sucrose, 10 mM Tris–HCl, 1 mM EDTA/Tris, pH 7.4), and pelleted at $600 \times g$ for 5 min. Mitochondria were washed three times by using mitochondrial wash buffer (10 mM Tris–HCl, pH 7.5, 2 mM EDTA, 0.5 mM sorbitol), and then collected at $7000 \times g$ for 20 min at 4 °C. Pellets were subsequently stored at −80 °C until use.

30 μg purified mitochondria were added to the activity assay buffer (27.5 mM $KH_2PO_4$ pH 7.4, 3.5 g/L BSA). Reactions were supplemented with DMSO, 1 μM antimycin, and pre-incubated at 30 °C for 10 min. The fumarate reduction reactions were initiated with a final concentration of 10 mM fumarate and 1 mM NADH. Samples were centrifuged and supernatants were analyzed by HPLC for succinate and fumarate.

### SA crystallization and recovery

SA crystals were recovered from acidic fermentation broth (Fig. 6b)[47]. The broth was centrifuged at $5000 \times g$ for 20 min to remove cell biomass and insoluble substances. Activated carbon (5%) was then added to bleach the broth under 30 °C and 250 rpm for 12 h. The clear Filtrate II was cooled to 2–4 °C for SA crystallization over 5 h. SA crystals I were gathered by centrifugation for 10 min at $5000 \times g$, 4 °C. Subsequently, Filtrate III was concentrated by vacuum distillation, and SA crystals II were collected through centrifugation. The final SA crystals were obtained by drying at 60 °C overnight, and their purity was measured by HPLC.

### Reporting summary

Further information on research design is available in the Nature Portfolio Reporting Summary linked to this article.

## Data availability

Genome sequence data of the evolved *Y. lipolytica* strains were available under the BioProject accession PRJNA998929. Source data are provided with this paper.

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

## Acknowledgements

This work was supported by the National Key R&D Program of China (grant no. 2021YFC2100500 to Q.Q.), the National Natural Science Foundation of China (grant no. 22208192 to Z.C.), the Major Scientific and Technological Innovation Project of Shandong Province (grant no. 2022CXGC020712-4 to Z.C.) and National Natural Science Foundation of Shandong Province (grant no. ZR2022ZD24 to J.H.). The authors thank the Analysis and Testing Center of SKLMT (State Key Laboratory of Microbial Technology, Shandong University), for providing facility assistance. The authors thank Suzhou Suzhen BIO-TECH Co., Ltd. for the support provided in scaled-up SA fermentation.

## Author contributions

Z.C., J.H., and Q.Q. conceived the study. Z.C. designed and performed most of the experiments. Q.W., J.H., and Q.Q. supervised the project. Z.S. performed the 13C-MFA experiments and analyzed the data. Y.Z., Z.J., and J.D. assisted with experimental performance. Z.C., J.N., J.H., and Q.Q. wrote and revised the manuscript.

## Competing interests

Z.C., J.H., and Q.Q. have filed two patents (CN201711276705.6 and CN202210395975.3) for protecting reductive TCA cycle-expressed *Y. lipolytica* strains for SA production. Other authors don't claim competing interests.
