## [Peer Review File · Nature Communications]

Reconfiguration of the reductive TCA cycle enables high-level succinic acid production by *Yarrowia lipolytica*Reviewers' Comments:

Reviewer #1:

Remarks to the Author:

General Comments:

This paper presents an original research on succinic acid (SA) production by a metabolically engineered yeast *Yarrowia lipolytica*. The following major findings/innovations have been achieved:

- 1) Using the acid-tolerant yeast *Yarrowia lipolytica* as the host to produce SA without pH control during the fermentation, which may benefit the downstream recovery and purification of SA.
- 2) Introducing the reductive TCA cycle into the cytosol of a previously engineered SDS strain for improved SA production yield.
- 3) Using an adaptive laboratory evolution (ALE) method to select a Pgl1-mutation strain, which redirects more carbon flow from PPP to glycolysis pathway for more NADH generation and SA production.
- 4) Localizing the major reductive TCA cycle steps in mitochondria and balancing the NADH generation and consumption for further improved SA production.
- 5) Demonstrating SA production at 50-L scale bioreactor for a titer of 119 g/L, a yield of 0.79 g/g glucose, and a rate of 1.79 g/L/h.

The results are impressive, and certainly are worthing of publication in some major journals in the field of biotechnology, bioengineering, synthetic biology and/or metabolic engineering. However, it is also my impression that the scientific significance may not be impactful enough to be published in *Nature Communications* as we do not see transformative innovations in the paper that may benefit the whole bioengineering/biotechnology community. Here are some of the major concerns:

- 1) It is not new that *Yarrowia lipolytica* is an acid-tolerant yeast, and has been previously used for SA production with glycerol as substrate (*Metab Eng* 42, 126-248 133, 2017).
- 2) Using reductive TCA cycle for SA production with improved yield is well known in the field. It has been verified in many other microorganisms, as mentioned by the authors in this paper and in the review (*Front. Bioeng. Biotechnol.* 10:843887. doi: 10.3389/fbioe.2022.843887).
- 3) The ALE method for selecting high-growth-rate and high-yield production strains is very common in the field. It is not convincing how the experiments were designed to achieve enough generations of strain growth that may lead to an effective mutation and evolution. It was also not mentioned why using gradually increased glucose concentration could serve as a selection pressure for evolving a strain for improved SA production.
- 4) There are other research in the field that have demonstrated better SA production titer and/or yield. For example, Ahn et al. demonstrated SA titer of 134 g/L, rate 21 g/L/h, and yield 0.81 g/g by using the strain *M. succiniciproducens* PALK (*Nat. Commun.* 11, 1970–1982. doi:10.1038/s41467-020-15839-z). Okino et al. demonstrated SA titer of 146 g/L, rate 3.17 g/L/h, and yield 0.92 g/g from glucose by using the strain *C. glutamicum* (*Appl. Microbiol. Biotechnol.* 81, 459–464. doi:10.1007/s00253-008-1668-y). There are also many other investigations on biosynthesis of SA and achieving very impressive TRYs. Although this paper used an acid-tolerant yeast, but the presented titer, rate, and yield were lower as compared to those previous research studies.

Other specific comments:

- 1) This paper really needs a Figure summarizes all detailed pathways and key genes that are related to PPP, glycolysis, TCA, and SA synthesis. The readers may feel hard to understand the roles of some genes/enzymes when they were mentioned for the first time in the paper. A table summarizes all the genes/enzymes, functions, sources, and references would also be helpful.
- 2) In the flask experiments, the shaking speed was set at 120 rpm. This may lead to low oxygen levels in flask. Through the paper mentioned that *Yarrowia lipolytica* is an aerobic strain, but the real fermentation condition may be conducted as microaerobic.
- 3) Similar to the question above, On page 6, ALE of engineered strain, explain why 120-rpm operation can ensure greater carbon flux into reductive TCA cycle? In general this makes sense due to lower

oxygen environment. Did the authors have any data to justify this?

4) Page 6, ALE of the strain, explain why most of the detected colonies showed increased SA production, but also produced more than 5 g/L malate as a by-product?

5) Are the initial glucose the total glucose added the same for all experiments shown in Fig 5? The protocols says "40-80 g/L" in the initial medium with 1-2 mL feeding of glucose (500 g/L).

6) For scaling from 5-L to 50-L bioreactor, what are the criteria for the scale-up? We see 400 rpm & 1 vvm for 5-L and 300 rpm & 0.5 vvm for 50-L, what parameters remain the same?

7) Was the dissolved oxygen level measured in all bioreactors run? Was it set to low levels so that reductive TCA cycle could be achieved?

8) A comparison between the SA/OD levels in flask, 5-L, and 50-L would be helpful for understanding the scale-up effect.

9) In ALE method, how much cells were transferred to the next round? How much time was needed for each round? How many generations were in each round of growth period and how many generations of growth were achieved at the end of the whole ALE process?

Reviewer #2:

Remarks to the Author:

The manuscript describes succinic acid production at low pH by strictly aerobic yeast *Yarrowia lipolytica* by implementing the combination of both oxidative and reductive TCA branches that allowed overcome the maximal theoretical yield of succinic acid generated via oxidative pathway only. Starting from the previously constructed SDH5 null mutant strain producing succinic acid via oxidative TCA, the authors first established the reductive TCA branch in cytosol by heterologous expression of soluble fumarate reductase, fumarase and endogenous malate dehydrogenase, which increased succinic acid yield but affected the growth. Adaptive laboratory evolution and genomic sequencing of the evolved strains revealed reduced flux via pentose phosphate pathway. The evolved strain was used to introduce a reductive TCA branch to the mitochondrial matrix. Fermentation in lab-scale and pilot-scale bioreactors was performed. The resultant strain produced up to 111.9 g/L of succinic acid without pH buffering in 50-L bioreactor with yield of 0.79 g/g and productivity of 1.79 g/L/h. Simple process of succinic acid purification and crystallization was demonstrated.

General comments:

Malate by-product produced by the evolved strains suggests the bottleneck at the fumarate reductase step. Did the authors test increasing the expression of the TbFrd gene encoding cytosolic enzyme (the second copy or stronger promoter)? Combination of mitochondrial and cytosolic TbFrd expression was advantageous compared to single copy cytosolic TbFrd expression. However, the author should compare multicopy Frd expression only in cytosol, multicopy Frd expression only in mitochondria and combination of expression in both compartments to suggest that relocation to mitochondria is beneficial.

It would be reasonable to test the effect of E.coli transhydrogenase expression in PGC91-rT strain before adaptive evolution leading to a decreased NADPH pool.

Fig 4b is not informative until the validation of the mitochondrial targeting sequence fused to mCherry has not been combined with a mitochondrial-specific dye and the merged images obtained.

Fig 4c should be corrected: cytosolic TbFrd strain, evolved cytosolic TbFrd strain, evolved cytosolic TbFrd strain together with mitochondrial Frd targeting.

Please specify whether the derivatives of evolved strain were cultivated in flasks at 120 rpm.

The authors stated the comparable process characteristics when fermenting in 50-L and 2000-L bioreactors, however, the results are presented only for 50-L.

Minor comments:

The full species names should be provided in figure caption (Extended Data Fig 3) and Supplementary Table 2, literature reference required for Supplementary Table 2 or text manuscript (Line 112).

The manuscript is generally written well but it is required double-check, e.g. Fig 2c is repeated twice,

Figure's link (Lines 282, 353) to be unified, proof-reading (Lines 424, 549), Table 3 (related to Fig 2), whether the reference 22 is correct (Line 91).

Table 4 should be completed with information about the developed strains for in vitro Pgl1 activity assay.

Reference to the paper(s) describing endogenous pyruvate decarboxylase bypass in *Yarrowia lipolytica* should be added.

Please add succinate and fumarate standards in Extended Data Fig 7.

Extended Data Fig 8: it is unclear why glucose spikes were added at different time points for Hi-SA0, Hi-SA1, and Hi-SA2 strains. Please, unify the cultivation conditions for all strains.

Reviewer #3:

Remarks to the Author:

In this manuscript, the strictly aerobic yeast *Yarrowia lipolytica* was engineered with the introduction of the reductive TCA cycle into both the cytosol and the mitochondria for efficient succinic acid production without pH control. The oxidative and reductive TCA cycle was coupled for NADH regeneration. In pilot-scale fermentation, the engineered strain produced 111.9 g/L SA with a yield of 0.79 g/g glucose within 62 h. It is interesting both for academic and industry.

1. Please give the full name when abbreviations like "Frd", "MTS" were used for the first time.

2. The effect of PCK on succinic acid production was much better than PYC in *E. coli*. How was the result in *Y. lipolytica*? Was there any difference for aerobic and anaerobic fermentation?

3. It seems that the function of fumarate reductase (Frd) was quite important in *Y. lipolytica*, which was much different from *E. coli*. Please give some explanation.

4. Fig. 1, "Data are presented as mean \pm s.e.m. (n = 2 of b-c, 3 of d-e biologically independent samples)". Normally there were at least three repeats in the experiment. Why were there only two in b-c?

5. Fig. 5, "Data are presented as mean \pm s.e.m. (n = 2 biologically independent samples)." Were there only two repeats?

6. The succinic acid titers for all PGC91-rTE strains were higher than PGC91-rT. Was there any special method for the selection of those strains, or only with cell growth restoring?

7. How were all the enzymes involved in the reductive TCA cycle overexpressed in THE mitochondria? Were those all with MTS?

8. It was found with ¹³C-Metabolic flux analysis that more than half flux into SA biosynthesis was derived from the reductive TCA cycle. Were there any inhibition effect of the enzymes of the reductive TCA cycle at aerobic condition?

9. The fermentation conditions were set as airflow rate of 1 vvm, stirring speed of 400 rpm in a 5-L bioreactor, and 300 rpm, 0.5 vvm in a 50-L bioreactor. How was the DO? Was the DO pattern optimized?

10. There were amino acids, which might be used as substrate for succinic acid synthesis, in YPD medium or modified minimal medium CM1 supplemented with corn steep powder. Was it adequate to calculate SA yield without modification?

11. How was the SA titer and yield if inorganic nitrogen source like ammonium salts, which were beneficial for SA separation and purification, were used?

Reviewer #1 (Remarks to the Author):

General Comments:

1. This paper presents an original research on succinic acid (SA) production by a metabolically engineered yeast *Yarrowia lipolytica*. The following major findings/innovations have been achieved:

1) Using the acid-tolerant yeast *Yarrowia lipolytica* as the host to produce SA without pH control during the fermentation, which may benefit the downstream recovery and purification of SA.

2) Introducing the reductive TCA cycle into the cytosol of a previously engineered SDS strain for improved SA production yield.

3) Using an adaptive laboratory evolution (ALE) method to select a Pgl1-mutation strain, which redirects more carbon flow from PPP to glycolysis pathway for more NADH generation and SA production.

4) Localizing the major reductive TCA cycle steps in mitochondria and balancing the NADH generation and consumption for further improved SA production.

5) Demonstrating SA production at 50-L scale bioreactor for a titer of 119 g/L, a yield of 0.79 g/g glucose, and a rate of 1.79 g/L/h.

The results are impressive, and certainly are worthing of publication in some major journals in the field of biotechnology, bioengineering, synthetic biology and/or metabolic engineering. However, it is also my impression that the scientific significance may not be impactful enough to be published in Nature Communications as we do not see transformative innovations in the paper that may benefit the whole

bioengineering/biotechnology community.

Response: Thanks for your valuable comments, which were essential for improving our manuscript. We believe that the major innovation of this study is the establishment of a reductive TCA pathway in mitochondria of aerobic yeast for efficient SA biosynthesis for the first time. Our results demonstrated that the mitochondrial localization of reductive TCA cycle is critical for improving SA production in *Yarrowia lipolytica*. Unlike facultative anaerobic yeasts which contain natural fermentative pathways and tend to accumulate reductive compounds under hypoxic conditions, *Y. lipolytica* have a high flux of oxidative TCA pathway and the introduction of the reductive SA production pathway in cytosol caused growth deficiency. When the reductive TCA cycle was localized to the mitochondria, NADH generated from the oxidative TCA cycle could be used for SA production via the reductive TCA cycle. This design can avoid the problem of NADH shortage caused by cytosol reductive TCA construction. The SA titer, yield and productivity of the engineered strain Hi-SA2 at low pH has reached the highest level reported, which has great industrial application prospects.

Here are some of the major concerns:

2. It is not new that *Yarrowia lipolytica* is an acid-tolerant yeast, and has been previously used for SA production with glycerol as substrate (Metab Eng 42, 126-248 133, 2017).

Response: Thanks for your comments. In the article you mentioned, our group constructed the oxidative SA producing strains of *Y. lipolytica* by disrupting succinate

dehydrogenase (SDH). Under the condition of high dissolved oxygen, the metabolic flux will enter the oxidative TCA cycle, which makes these engineered *Y. lipolytica* strains exhibit excellent SA production performance. However, the oxidative branch of TCA cycle releases a large amount of CO₂, the SA yield was only 0.53 g/g glycerol. Furthermore, the glucose consumption rate in the SDH inactivated strains was significantly lower than that of glycerol, and the SA production was low when glucose was used as the sole carbon source. Although glycerol is a byproduct of biodiesel, its supply is insufficient and its price fluctuates greatly. Considering the stability of the supply chain, glucose is a more suitable feedstock for large-scale SA production. Therefore, we further engineered the strains to improve the yield using glucose as feedstock in this study.

3. Using reductive TCA cycle for SA production with improved yield is well known in the field. It has been verified in many other microorganisms, as mentioned by the authors in this paper and in the review (Front. Bioeng. Biotechnol. 10:843887. doi: 10.3389/fbioe.2022.843887).

Response: We agree that using reductive TCA cycle for SA production is a well-known approach to improve SA yield. In nature, only some strict or facultative anaerobic microorganisms have reductive TCA cycle. Most aerobic organisms, including *Y. lipolytica*, lack the reductive SA synthetic pathway. The construction of reductive TCA cycle has successfully improved SA production in many bacteria. For acid tolerance yeasts, although some studies have shown that heterologous reductive TCA can be

expressed in some ethanol-producing yeasts (*Saccharomyces cerevisiae*, *Issatchenkia orientalis*), their SA yield is still unsatisfactory. When using strictly aerobic yeast *Y. lipolytica* as a chassis to construct the reductive TCA cycle, it is also challenging to maintain the redox balance. Different from bacteria, yeasts have subcellular compartment and their cofactor cannot transport freely. Cytosolic cofactor imbalance restricts the improvement of SA yield. This study revealed that the mitochondrial matrix is the optimal location for reductive TCA cycle, and *Y. lipolytica* can be used to construct cell factory for efficient SA biosynthesis, which has important theoretical and practical significance.

4. The ALE method for selecting high-growth-rate and high-yield production strains is very common in the field. It is not convincing how the experiments were designed to achieve enough generations of strain growth that may lead to an effective mutation and evolution. It was also not mentioned why using gradually increased glucose concentration could serve as a selection pressure for evolving a strain for improved SA production.

Response: As the reviewer mentioned, adaptive laboratory evolution (ALE) is an effective method to evolve strains to improve the growth. In this study, the SDH-deficient *Y. lipolytica* strain was used as the original strain, which has impaired glucose metabolism. Once the reductive TCA pathway was introduced, engineered strain PGC91-rT was almost unable to grow in glucose medium. To restore cell growth, PGC91-rT strain was subjected to ALE with gradually increased glucose concentration

as described in the references ¹⁻⁴. Our selection pressure is cell growth. After about 40 times transfers, the strains with better growth rate were obtained. We monitored the cell growth during evolution and the increased growth rate can be used as an evidence for an effective mutation and evolution. We found the evolutionary strains with better growth also had higher SA production.

In our design, glucose concentration was gradually increased so that the strain can adapt high glucose concentration for rapid glucose metabolism and SA generation. In our previous experiment, we observed that the cell growth is lower when glucose concentration is high, and therefore gradually increased glucose concentration were used in ALE. We have edited the manuscript to include more information of ALE in methods.

[References]

- Yang, X. F., Wang, H. M., Li, C., and Lin, C. S. K. (2017) Restoring of glucose metabolism of engineered *Yarrowia lipolytica* for succinic acid production via a simple and efficient adaptive evolution strategy, *J Agr Food Chem* 65, 4133-4139.
- Jiang, L., Li, S., Hu, Y., Xu, Q., and Huang, H. (2012) Adaptive evolution for fast growth on glucose and the effects on the regulation of glucose transport system in *Clostridium tyrobutyricum*, *Biotechnol Bioeng* 109, 708-718.
- Yuzbashev, T. V., Yuzbasheva, E. Y., Sobolevskaya, T. I., Laptev, I. A., Vybornaya, T. V., Larina, A. S., Matsui, K., Fukui, K., and Sineoky, S. P. (2010) Production of succinic acid at low pH by a recombinant strain of the aerobic yeast *Yarrowia lipolytica*, *Biotechnol Bioeng* 107, 673-682.
- Yuzbashev, T. V., Bondarenko, P. Y., Sobolevskaya, T. I., Yuzbasheva, E. Y., Laptev, I. A., Kachala, V. V., Fedorov, A. S., Vybornaya, T. V., Larina, A. S., and Sineoky, S. P. (2016) Metabolic evolution and ¹³C flux analysis of a succinate dehydrogenase

deficient strain of *Yarrowia lipolytica*, *Biotechnol Bioeng* 113, 2425-2432.

5. There are other research in the field that have demonstrated better SA production titer and/or yield. For example, Ahn et al. demonstrated SA titer of 134 g/L, rate 21 g/L/h, and yield 0.81 g/g by using the strain *M. succiniciproducens* PALK (Nat. Commun. 11, 1970–1982. doi:10.1038/s41467-020-15839-z). Okino et al. demonstrated SA titer of 146 g/L, rate 3.17 g/L/h, and yield 0.92 g/g from glucose by using the strain *C. glutamicum* (Appl. Microbiol. Biotechnol. 81, 459–464. doi:10.1007/s00253-008-1668-y). There are also many other investigations on biosynthesis of SA and achieving very impressive TRYs. Although this paper used an acid-tolerant yeast, but the presented titer, rate, and yield were lower as compared to those previous research studies.

Response: Thanks for your kindly suggestion. SA is chemical with diverse applications. Many efforts have been devoted to construct and optimize microbial cell factories for low-cost production of SA. As the reviewer mentioned, some studies have achieved impressive TRYs in bacteria. SA production using bacteria indeed has many advantages, such as fast growth rate, high TRYs, etc., and bacteria has also been used for industrial production of SA. However, a large amount of NaOH or Ca(OH)₂ is added to maintain neutral pH during bacterial fermentation, and an equal amount of sulfuric acid is needed to obtain free SA during product separation, which undoubtedly increases the production cost of SA.

Yeasts have excellent acid tolerance and could achieve low pH (< 3) SA fermentation,

but poor SA production capacity limits their application. In this work, without adjusting the pH, the SA titer, yield and productivity of the engineered *Y. lipolytica* strain reached 111.9 g/L, 79 g/g glucose and 1.79 g/L/h, respectively. In collaboration with the Suzhou Suzhen BIO-TECH Co., Ltd., we evaluated the production costs of the whole process (raw materials, energy, labor, extraction, etc.) of low pH SA fermentation of *Y. lipolytica*. Comparing with petrochemical-based SA, the cost is competitive. It is also worth noting that acidic fermentation broth containing high concentrations of SA is highly cytotoxic, and it is more difficult to increase the SA production at low pH. According to literature reports, the SA production of SDH-deficient *Y. lipolytica* could reach nearly 200 g/L when the pH was adjusted at 6.5. Therefore, we believe that the engineered strains constructed here can also exhibit higher SA production if pH was adjusted.

Other specific comments:

6. This paper really needs a Figure summarizes all detailed pathways and key genes that are related to PPP, glycolysis, TCA, and SA synthesis. The readers may feel hard to understand the roles of some genes/enzymes when they were mentioned for the first time in the paper. A table summarizes all the genes/enzymes, functions, sources, and references would also be helpful.

Response: Thanks for your kindly suggestion, we have added a figure that included all detailed metabolic pathway to the Supplementary material to make it easier for readers to understand the role of related genes/enzymes (Extended Data Fig. 1). In addition, we

list all the genes/enzymes, functions, sources, and references in Supplementary Tab. 4.

Extended Data Fig. 1 Schematic illustration of all SA biosynthesis related metabolic pathway in *Yarrowia lipolytica*. Glucose transport: hexose transporter (Yht); Pentose phosphate pathway: 6-phosphate-glucose dehydrogenase (Zwf); 6-phospho-gluconolactonase (Pgl); 6-phospho-gluconate dehydrogenase (Pgd); Oxidative TCA pathway: pyruvate dehydrogenase (Pdh); citrate synthetase (Cit); aconitate hydratase (Aco); α -ketoglutarate dehydrogenase (Kgdh); succinyl-CoA synthetase (Scs); Reductive TCA pathway: pyruvate carboxylase (Pyc); malate dehydrogenase (Mdh); fumarate hydratase (Fum); fumarate reductase (Frd); mitochondrial fumarate reductase (mFrd); Glyoxylate bypass: isocitrate lyase (Icl); malate synthetase (Mls); SA transport: dicarboxylic acid transporter (Mae).

7. In the flask experiments, the shaking speed was set at 120 rpm. This may lead to low oxygen levels in flask. Through the paper mentioned that *Yarrowia lipolytica* is an aerobic strain, but the real fermentation condition may be conducted as microaerobic.

Response: Thanks for your kindly suggestion. Although *Y. lipolytica* is an aerobic strain, it is capable to grow under low oxygen levels. Under microaerobic conditions, more metabolic flux can enter the reductive TCA cycle, so it is more suitable for SA fermentation. We have edited the manuscript to emphasize the role of microaerobic conditions for SA fermentation.

8. Similar to the question above, On page 6, ALE of engineered strain, explain why 120-rpm operation can ensure greater carbon flux into reductive TCA cycle? In general this makes sense due to lower oxygen environment. Did the authors have any data to justify this?

Response: As shown in Extended Data Fig. 2b, the SA yield of PGC91-TbFrd increased as the rotational speed decreased, which indicates that low oxygen condition is beneficial for the reductive TCA cycle. In addition, we compared the metabolic flux distribution of the engineered strain Hi-SA2 at 220 rpm and 120 rpm, and found higher carbon flux into reductive TCA cycle at lower rotational speed (Extended Data Fig. 10).

Extended Data Fig. 2 Comparison of SA production performance between PGC91 (a) and PGC91-TbFrd (b) strains in YPD media with different shaking speeds. Data are presented as mean \pm s.e.m. (n = 3 biologically independent samples).

Extended Data Fig. 10 Flux distributions from ¹³C-Metabolic Flux Analysis. Two flux values are listed for the engineered strain Hi-SA2 cultivated at 220 rpm (top, black font) and 120 rpm (bottom, red font).

9. Page 6, ALE of the strain, explain why most of the detected colonies showed increased SA production, but also produced more than 5 g/L malate as a by-product?

Response: The growth of the evolved strains was restored, and higher biomass and sugar consumption rates were accompanied by improved SA production. It is speculated that glycolytic flux has been improved in evolved strains, while the gene expression or enzyme activity of fumarate reductase in reductive TCA cycle was insufficient, leading to the accumulation of precursor malate.

10. Are the initial glucose the total glucose added the same for all experiments shown in Fig 5? The protocols says “40-80 g/L” in the initial medium with 1-2 mL feeding of glucose (500 g/L).

Response: For all the SA fermentation in Fig. 5, initial concentration of glucose was 80 g/L. We have added the initial concentration of glucose in the figure legend.

11. For scaling from 5-L to 50-L bioreactor, what are the criteria for the scale-up? We see 400 rpm & 1 vvm for 5-L and 300 rpm & 0.5 vvm for 50-L, what parameters remain the same?

Response: As mentioned above, appropriate dissolved oxygen conditions are essential for efficient SA production of reductive TCA cycle overexpressed strains. We optimized the fermentation conditions, and the speed and air flow rate were used as the main parameters to maintain low dissolved oxygen level ($\leq 10\%$). The fermentation conditions in 5-L and 50-L are different because they are optimized separately.

12. Was the dissolved oxygen level measured in all bioreactors run? Was it set to low levels so that reductive TCA cycle could be achieved?

Response: Yes, the dissolved oxygen was measured in the bioreactors, and it is controlled at low level. The data related to dissolved oxygen level were added in Extended Data Fig. 10 and Extended Data Fig. 13. In the 5-L bioreactors, the dissolved oxygen level decreased rapidly after inoculation and then maintained at a low level.

Similar trend was observed in 50-L fermentation.

Extended Data Fig. 13 Fed-batch fermentation of Hi-SA2 strain in 5-L bioreactor with the media of modified CM1. Data are presented as mean \pm s.e.m. (n = 2 biologically independent samples).

13. A comparison between the SA/OD levels in flask, 5-L, and 50-L would be helpful for understanding the scale-up effect.

Response: Thanks for your kindly suggestion, SA/OD levels of Hi-SA2 strain in flask, 5-L, and 50-L were 7.82, 2.45, and 2.78, respectively. It seems that in the scale-up fermentation, the biomass production is higher, therefore the SA production capacity per unit biomass in the scale-up fermentation is decreasing.

14. In ALE method, how much cells were transferred to the next round? How much time was needed for each round? How many generations were in each round of growth period and how many generations of growth were achieved at the end of the whole ALE process?

Response: In ALE, the initial OD₆₀₀ of each round is about 0.5. The cultivation times needed for each round were about 36-48 h. In addition to generations, the number of

transfers is also a commonly used measure of ALE. Cell growth rates during evolution are constantly changing, making it difficult to accurately measure generations. We use transfer number and did not count the generations for the whole ALE process.

Reviewer #2 (Remarks to the Author):

The manuscript describes succinic acid production at low pH by strictly aerobic yeast *Yarrowia lipolytica* by implementing the combination of both oxidative and reductive TCA branches that allowed overcome the maximal theoretical yield of succinic acid generated via oxidative pathway only. Starting from the previously constructed SDH5 null mutant strain producing succinic acid via oxidative TCA, the authors first established the reductive TCA branch in cytosol by heterologous expression of soluble fumarate reductase, fumarase and endogenous malate dehydrogenase, which increased succinic acid yield but affected the growth. Adaptive laboratory evolution and genomic sequencing of the evolved strains revealed reduced flux via pentose phosphate pathway. The evolved strain was used to introduce a reductive TCA branch to the mitochondrial matrix. Fermentation in lab-scale and pilot-scale bioreactors was performed. The resultant strain produced up to 111.9 g/L of succinic acid without pH buffering in 50-L bioreactor with yield of 0.79 g/g and productivity of 1.79 g/L/h. Simple process of succinic acid purification and crystallization was demonstrated.

General comments:

15. Malate by-product produced by the evolved strains suggests the bottleneck at the fumarate reductase step. Did the authors test increasing the expression of the TbFrd

gene encoding cytosolic enzyme (the second copy or stronger promoter)? Combination of mitochondrial and cytosolic TbFrd expression was advantageous compared to single copy cytosolic TbFrd expression. However, the author should compare multicopy Frd expression only in cytosol, multicopy Frd expression only in mitochondria and combination of expression in both compartments to suggest that relocation to mitochondria is beneficial.

Response: Thanks for your kindly suggestion. As shown in Fig. 4c, cytosolic and mitochondrial TbFrd has been expressed in PGC91-rTE1-1 (the strain with one copy of cytosolic TbFrd), respectively. All the expressed strains no longer accumulated the by-product malate (data not shown). Mitochondrial TbFrd expression (PGC91-rTE1-mTbFrd) produced 30.8 g/L SA, which is 45.4% higher than that of PGC91-rTE1-1. In contrast, expression of one more copy of cytosolic TbFrd caused a decrease in SA titer and OD₆₀₀ in the control strain. These results demonstrated that relocation to mitochondria is beneficial for SA production. We have added the comparison of cytosolic Frd expression and mitochondria Frd expression in the manuscript (Page 11, Line 255).

Fig. 4c Expression of cytosolic and mitochondrial soluble Frd in the evolved strain PGC91-rTE1-1 to enhance SA production.

16. It would be reasonable to test the effect of *E.coli* transhydrogenase expression in PGC91-rT strain before adaptive evolution leading to a decreased NADPH pool.

Response: Thanks for your valuable comments. As the reviewer suggested, transhydrogenase derived from *E. coli* had been expressed in PGC91-rT and PGC91-rTE-1 strain (as shown below). However, the titer of SA was not significantly increased. We tested different strategies to increase the NADH pool in cytosol, but the SA production is still relatively low (Extended Data Fig. 6).

Fig. (1) Increasing cytoplasmic NADH supply by transhydrogenase overexpression for improved SA production of *Y. lipolytica*.

Extended Data Fig. 6 Increasing cytoplasmic NADH supply for SA synthesis. **a**. Schematic diagram of metabolic engineering strategies to improve cytoplasmic NADH regeneration in *Y. lipolytica*; **b**. Effects of transhydrogenase and pyruvate decarboxylase bypass overexpression on SA production of PGC91-rTE1-1 strain. The initial concentration of glucose was 60 g/L. Data are presented as mean \pm s.e.m. ($n = 2$ biologically independent samples). *EcSthA* encoding *E. coil* transhydrogenase, *YIPdc2* encoding pyruvate decarboxylase, *YIald5* encoding aldehyde dehydrogenase, *YIAcs1* encoding endogenous acetyl-CoA synthetase, *SeAcs* encoding *Salmonella enteric* acetyl-CoA synthetase, *YIMIs* encoding malate synthetase, *YIic11* encoding isocitrate lyase.

17. Fig 4b is not informative until the validation of the mitochondrial targeting sequence fused to mCherry has not been combined with a mitochondrial-specific dye and the merged images obtained.

Response: Thanks for your kindly suggestion, we have tested the subcellular localization of the mitochondrial targeting sequence (MTS) several times.

In our previous work, Mito-Tracker Red CMXRos (Beyotime, ShangHai, China) was successfully used for mitochondrial staining. However, when we tried to fuse MTS of cytochrome c oxidase 5b to green fluorescent protein hrGFP, the green fluorescence can't be detected. Although fluorescence colocalization data were not available, the fluorescence distribution of MTS-mCherry was different from that of cytosolic mCherry.

Recently, we used the Mito-Tracker Green (Beyotime, ShangHai, China) to stain the mitochondria and MTS was fused with mCherry. As shown below, we found that the living cells cannot be stained by mitochondrial fluorescent dye.

Fig. (2) Fluorescence microscopy of yeast strains expressing cytoplasmic and mitochondrial mCherry and stained with Mito-Tracker Red CMXRos. WT stands for the wild type strain Po1f. Red fluorescent protein mCherry with and without mitochondrial targeting sequence were expressed in Po1f strain, respectively.

To proving the mitochondrial localization, we had purified the mitochondria of PGC91-rTE and PGC91-rTE-mTbFrd, and detected the Frd activity. Only the mTbFrd-overexpressing strain was able to produce SA using fumarate and NADH the substrates,

confirming the functional expression of TbFrd in mitochondria (Extended Data Fig. 8). The fermentation results of mitochondrial localization were different from cytosolic localization. Mitochondrial localization of TbFrd significantly improved the SA synthesis ability of PGC91-rTE1-1, and its titer and yield of SA reached 30.8 g/L and 0.83 g/g glucose, respectively (Fig. 4c). In contrast, the expression of another copy of cytosolic TbFrd in PGC91-rTE1-1 strain caused a decrease in SA titer and OD₆₀₀. Based on this result, we can conclude that MTS can play a role in mitochondrial localization.

Extended Data Fig. 8 Fumarate reductase activity in purified mitochondria from PGC91-rTE1-1 and PGC91-rTE-mTbFrd cells. The fumarate reduction reaction was initiated with 10 mM fumarate and 1 mM NADH, and monitored through the production of succinate over time. 10 mg/L succinate and 1 g/L fumarate were used as standards for HPLC detection.

18. Fig 4c should be corrected: cytosolic TbFrd strain, evolved cytosolic TbFrd strain, evolved cytosolic TbFrd strain together with mitochondrial Frd targeting.

Answer: Thanks for your kindly suggestion, the figure notes of Fig 4c has been modified to avoid confusion.

19. Please specify whether the derivatives of evolved strain were cultivated in flasks at 120 rpm.

Response: All shaking flasks fermentations were performed at 120 rpm unless otherwise stated.

20. The authors stated the comparable process characteristics when fermenting in 50-L and 2000-L bioreactors, however, the results are presented only for 50-L.

Answer: Thanks for your kindly suggestion, the industrial scale SA fermentation in 2000-L bioreactor was performed by Suzhou Suzhen BIO-TECH Co., Ltd. The relevant data are not presented to avoid potential commercial conflicts of interest.

Minor comments:

21. The full species names should be provided in figure caption (Extended Data Fig 3) and Supplementary Table 2, literature reference required for Supplementary Table 2 or text manuscript (Line 112).

Response: The full species names have been provided in Extended Data Fig. 4 and Supplementary Table 2. References were added in text manuscript.

22. The manuscript is generally written well but it is required double-check, e.g. Fig 2c is repeated twice, Figure's link (Lines 282, 353) to be unified, proof-reading (Lines 424, 549), Table 3 (related to Fig 2), whether the reference 22 is correct (Line 91).

Response: Thanks for your reminding, these errors have been corrected and we have carefully checked the full text to avoid any typing errors.

23. Table 4 should be completed with information about the developed strains for in vitro Pgl1 activity assay.

Response: The developed strains for in vitro Pgl1 activity assay have been added in Supplementary Tab. 5.

24. Reference to the paper(s) describing endogenous pyruvate decarboxylase bypass in *Yarrowia lipolytica* should be added.

Response: The papers describing endogenous pyruvate decarboxylase bypass in *Y. lipolytica* have been referenced in Page 9, Line 218.

25. Please add succinate and fumarate standards in Extended Data Fig 7.

Answer: The succinate and fumarate standards have been added in Extended Data Fig 8.

Extended Data Fig. 8 Fumarate reductase activity in purified mitochondria from

PGC91-rTE1-1 and PGC91-rTE-mTbFrd cells. The fumarate reduction reaction was initiated with 10 mM fumarate and 1 mM NADH, and monitored through the production of succinate over time. 10 mg/L succinate and 1 g/L fumarate were used as standards for HPLC detection.

26. Extended Data Fig 8: it is unclear why glucose spikes were added at different time points for Hi-SA0, Hi-SA1, and Hi-SA2 strains. Please, unify the cultivation conditions for all strains.

Response: These three engineered strains had different glucose consumption rates, so it was difficult to keep the feeding time consistent. In general, the SA fermentation in shaking flasks were relatively stable, the glucose feeding time will not affect the production of SA. The cultivation conditions were added in the legend of Extended Data Fig 8.

Reviewer #3 (Remarks to the Author):

27. In this manuscript, the strictly aerobic yeast *Yarrowia lipolytica* was engineered with the introduction of the reductive TCA cycle into both the cytosol and the mitochondria for efficient succinic acid production without pH control. The oxidative and reductive TCA cycle was coupled for NADH regeneration. In pilot-scale fermentation, the engineered strain produced 111.9 g/L SA with a yield of 0.79 g/g glucose within 62 h. It is interesting both for academic and industry.

Response: Thank you for your positive evaluation to work.

28. Please give the full name when abbreviations like “Frd”, “MTS” were used for the first time.

Response: Full names of abbreviations have been provided for the first time in manuscript.

29. The effect of PCK on succinic acid production was much better than PYC in *E. coli*. How was the result in *Y. lipolytica*? Was there any difference for aerobic and anaerobic fermentation?

Response: As described in our previous works, the expression of PCK and PYC can both improve the SA biosynthesis of *Y. lipolytica*. However, when the TbFrd was expressed in *Y. lipolytica* PGC62 (*MatA*, *xpr2-322*, *axp-2*, *leu2-270*, *ura3-302*, Δ *Sdh5::loxP*, Δ *Ach1::loxP*, *ScPck*) strain, the SA yield was only 0.51 g/g glucose. These results were shown in the references ^{5,6}. In this study, the expression of TbFrd in PGC91 (*MatA*, *xpr2-322*, *axp-2*, *leu2-270*, *ura3-302*, Δ *Sdh5::loxP*, Δ *Ach1::loxP*, *YlPyc*) resulted in the highest SA yield of 0.62 g/g glucose (Fig. 1b). Thus, PYC is more suitable for the construction of reductive TCA pathway in *Y. lipolytica*. As *Y. lipolytica* is a strictly aerobic organism and cannot grow under anaerobic conditions, only aerobic or microaerobic SA fermentation was investigated in this study. In general, lower dissolved oxygen results in a higher SA yield.

[References]

Cui, Z. Y., Gao, C. J., Li, J. J., Hou, J., Lin, C. S. K., and Qi, Q. S. (2017) Engineering

of unconventional yeast *Yarrowia lipolytica* for efficient succinic acid production from glycerol at low pH, *Metab Eng* 42, 126-133.

Jiang, Z. N., Cui, Z. Y., Zhu, Z. W., Liu, Y. H., Tang, Y. J., Hou, J., and Qi, Q. S. (2021) Engineering of *Yarrowia lipolytica* transporters for high-efficient production of biobased succinic acid from glucose, *Biotechnol Biofuels* 14.

30. It seems that the function of fumarate reductase (Frd) was quite important in *Y. lipolytica*, which was much different from *E. coli*. Please give some explanation.

Response: Fumarate reductases (Frds) is a key enzyme in the reductive TCA cycle, and catalyzes the reduction of fumarate to SA. In *E. coli*, an endogenous Frd complex is present in the cell membrane, which can accept electrons from complex I of the electron transport chain for fumarate reducing. However, due to the lack of endogenous Frds, many eukaryotic microorganisms (including *Y. lipolytica*) are unable generate SA via the reductive TCA branch. In this study, the soluble Frd from protozoans that uses NADH as a cofactor was used to construct reductive TCA cycle in *Y. lipolytica*.

31. Fig. 1, “Data are presented as mean \pm s.e.m. (n = 2 of b-c, 3 of d-e biologically independent samples)”. Normally there were at least three repeats in the experiment.

Why were there only two in b-c?

Response: Random genome integration approach was used for the gene overexpression in this work. After transformation, ten yeast colonies were first picked to determine the SA titer and yield by shaking flasks fermentation. Because we have confirmed the SA

production of the recombinant strains in first round of fermentation, in the second round, only two replicates were used for data analysis.

32. Fig. 5, “Data are presented as mean \pm s.e.m. (n = 2 biologically independent samples).” Were there only two repeats?

Answer: The reason is same as the above question.

33. The succinic acid titers for all PGC91-rTE strains were higher than PGC91-rT. Was there any special method for the selection of those strains, or only with cell growth restoring?

Response: The restoration of cell growth during adaptive evolution was the only indicator for selecting evolved strains. The strains with recovered cell growth had better glucose metabolism, thereby promote SA production.

34. How were all the enzymes involved in the reductive TCA cycle overexpressed in THE mitochondria? Were those all with MTS?

Response: The subcellular localization of enzymes involved in reductive TCA cycle were first analyzed by online tool Softberry (<http://www.softberry.com/berry.phtml>) and MITOPROT (<https://ihg.gsf.de/ihg/mitoprot.html>). For cytoplasmic proteins, MTS was added to their N-terminus of the proteins to achieve mitochondrial relocation. For the enzymes which itself has MTS, we did not add addition MTS and just expressed the original sequences of the enzymes. These descriptions have been added in methods

section.

35. It was found with ¹³C-Metabolic flux analysis that more than half flux into SA biosynthesis was derived from the reductive TCA cycle. Were there any inhibition effect of the enzymes of the reductive TCA cycle at aerobic condition?

Response: As shown in modified Extended Data Fig. 10, the ratio of metabolic flux between reductive and oxidative TCA cycles is affected by dissolved oxygen level. At high speed of 220 rpm, only about 30 % of the carbon flux into SA biosynthesis was derived from the reductive TCA cycle, while more than half flux into SA biosynthesis was derived from the reductive TCA cycle when the rotational speed was 120 rpm. We believe oxygen level is the main inhibition effect to the flux to reductive TCA cycle, as it changes the redox state and respiration.

Extended Data Fig. 10 Flux distributions from ¹³C-Metabolic Flux Analysis. Two flux values are listed for the engineered strain Hi-SA2 cultivated at 220 rpm (top, black font) and 120 rpm (bottom, red font).

36. The fermentation conditions were set as airflow rate of 1 vvm, stirring speed of 400 rpm in a 5-L bioreactor, and 300 rpm, 0.5 vvm in a 50-L bioreactor. How was the DO? Was the DO pattern optimized?

Response: The DO was monitored during the fermentation and it was added in Extended Data Fig. 11 and Extended Data Fig.13. For the scale-up fermentation in bioreactors, the rotational speed and airflow rate were optimized to obtain optimal SA production. The DO pattern was not optimized.

37. There were amino acids, which might be used as substrate for succinic acid synthesis, in YPD medium or modified minimal medium CM1 supplemented with corn steep powder. Was it adequate to calculate SA yield without modification?

Response: The amino acids present in complex nitrogen sources such as yeast extract, tryptone, and corn steep powder do have an impact on SA yield calculations. For example, the SA yield of Hi-SA2 strain in 5L bioreactor with YPD medium was significantly higher than that with modified CM1 medium. Even though, modified CM1 medium supplemented with only a small amount of corn steep powder (6 g/L). It contains about 41% protein, and the final SA titer up to 110 g/L, therefore its effect on SA yield is limited.

38. How was the SA titer and yield if inorganic nitrogen source like ammonium salts, which were beneficial for SA separation and purification, were used?

Response: Thank you for your valuable comment. Although the use of inorganic nitrogen source will be beneficial for SA separation and purification, a large amount of inorganic salt added during industrial production will result in an increase in the cost of production. Corn steep powder is a by-product of cornstarch production, which is relative cheap and also rich in soluble proteins, trace elements and some precursors, and has been widely used in large-scale microbial fermentation as cheap source of nitrogen. Therefore, we did not used inorganic nitrogen source.

References:

- [1] Yang, X. F., Wang, H. M., Li, C., and Lin, C. S. K. (2017) Restoring of glucose metabolism of engineered *Yarrowia lipolytica* for succinic acid production via a simple and efficient adaptive evolution strategy, *J Agr Food Chem* 65, 4133-4139.
- [2] Jiang, L., Li, S., Hu, Y., Xu, Q., and Huang, H. (2012) Adaptive evolution for fast growth on glucose and the effects on the regulation of glucose transport system in *Clostridium tyrobutyricum*, *Biotechnol Bioeng* 109, 708-718.
- [3] Yuzbashev, T. V., Yuzbasheva, E. Y., Sobolevskaya, T. I., Laptev, I. A., Vybornaya, T. V., Larina, A. S., Matsui, K., Fukui, K., and Sineoky, S. P. (2010) Production of succinic acid at low pH by a recombinant strain of the aerobic yeast *Yarrowia lipolytica*, *Biotechnol Bioeng* 107, 673-682.
- [4] Yuzbashev, T. V., Bondarenko, P. Y., Sobolevskaya, T. I., Yuzbasheva, E. Y., Laptev, I. A., Kachala, V. V., Fedorov, A. S., Vybornaya, T. V., Larina, A. S., and Sineoky, S. P. (2016) Metabolic evolution and ¹³C flux analysis of a succinate

dehydrogenase deficient strain of *Yarrowia lipolytica*, *Biotechnol Bioeng* 113, 2425-2432.

- [5] Cui, Z. Y., Gao, C. J., Li, J. J., Hou, J., Lin, C. S. K., and Qi, Q. S. (2017) Engineering of unconventional yeast *Yarrowia lipolytica* for efficient succinic acid production from glycerol at low pH, *Metab Eng* 42, 126-133.
- [6] Jiang, Z. N., Cui, Z. Y., Zhu, Z. W., Liu, Y. H., Tang, Y. J., Hou, J., and Qi, Q. S. (2021) Engineering of *Yarrowia lipolytica* transporters for high-efficient production of biobased succinic acid from glucose, *Biotechnol Biofuels* 14.

Reviewers' Comments:

Reviewer #1:

Remarks to the Author:

General Comment:

As the reviewer #1, I appreciate the authors' great efforts in addressing the concerns that I raised in the first round of review. After I further reviewed the responses and the revised paper, I think most the questions/concerns have been addressed well. Although I still have the conservative concern of the novelty of the paper as it mainly combines the findings from the previous research by the authors and others in the strain *Yarrowia lipolytica*, I would respect and follow the decision/recommendation by the Editor and other two reviewers.

Specific comments:

(1) The authors mentioned in the response to the comment #5 "In collaboration with the Suzhou Suzhen BIO-TECH Co., Ltd., we evaluated the production costs of the whole process (raw materials, energy, labor, extraction, etc.) of low pH SA fermentation of *Y. lipolytica*. Comparing with petrochemical-based SA, the cost is competitive." It would be nice to include some details (specific numbers) in the discussion of the revised paper.

(2) The generation number for the ALE process can be easily estimated based on the OD change in each growth transfer. For example, if the OD increased from 0.5 to 15, that's a number close to 2^5 , then it is about 5 generations. For a total of 40 transfers with each having similar growth period, then a total of $5 \times 40 = 80$ generations was estimated. Using the number of transfers, although is a common practice, but is not a scientifically meaningful. It would be nice to have the estimated generations included.

(3) In the response to the comment #13, the authors indicated that "SA/OD levels of Hi-SA2 strain in flask, 5-L, and 50-L were 7.82, 2.45, and 2.78, respectively. It seems that in the scale-up fermentation, the biomass production is higher, therefore the SA production capacity per unit biomass in the scale-up fermentation is decreasing". It would be nice to include this in the discussion to show the potential of further improvement opportunity via fermentation optimization and scale-up in future.

Reviewer #2:

Remarks to the Author:

The manuscript has been carefully revised. Most comments have been addressed carefully. However, there are minor points:

1. Please mention in the text that PGC91-rTE1-1 derivatives harboring additional TbFrd overexpression no longer accumulated by-product malate.

2. The experiment with mCherry localization is not informative. Based on the results, the only conclusion that the authors can have is that "the fluorescence distribution of MTS-mCherry was different from that of cytosolic mCherry". This does not support the statement in the text "The fluorescence distribution of MTS-mCherry was consistent with mitochondrial localization, while mCherry without the signal peptide showed a cytosolic distribution (Fig. 4b)." I would suggest either remove all information about mCherry targeting from the manuscript or move it to Supplementary Materials with correct conclusions.

3. Line 68: repetition of the noun cycle.

Reviewer #3:

Remarks to the Author:

The authors have modified the manuscript carefully. I believe it is ready for publication.

We thank all the reviewers for their valuable comments and suggestions. We have edited the manuscript carefully according to the reviewers' thoughtful comments and suggestions, and responded to these comments point-by-point. All changes made to the text are labeled in red.

Reviewer #1 (Remarks to the Author):

General Comment:

As the reviewer #1, I appreciate the authors' great efforts in addressing the concerns that I raised in the first round of review. After I further reviewed the responses and the revised paper, I think most the questions/concerns have been addressed well. Although I still have the conservative concern of the novelty of the paper as it mainly combines the findings from the previous research by the authors and others in the strain *Yarrowia lipolytica*, I would respect and follow the decision/recommendation by the Editor and other two reviewers.

Response: We sincerely appreciate your valuable suggestions and comments, which are very helpful to improve the quality of our manuscript.

Specific comments:

(1) The authors mentioned in the response to the comment #5 "In collaboration with the Suzhou Suzhen BIO-TECH Co., Ltd., we evaluated the production costs of the whole process (raw materials, energy, labor, extraction, etc.) of low pH SA fermentation of *Y. lipolytica*. Comparing with petrochemical-based SA, the cost is competitive." It would

be nice to include some details (specific numbers) in the discussion of the revised paper.

Response: Thanks for your comments. The mainly feedstock cost for bio-SA production is from glucose. The glucose price maintains about ¥4.0/kg. When the SA yield is 0.8 g/g glucose, the feedstock cost is about ¥5.0 to produce 1 kg SA. The purification is around ¥3.0/kg. Other cost during fermentation and purification is about ¥3.0-4.0/kg. The production cost of low pH SA fermentation is ¥11-12/kg, which was consistently below the petrochemical-based SA market price range of ¥14-18/kg.

At present, cost estimation of low pH SA production is only the preliminary result, and therefore we do not think it appropriate to discuss it in the revised manuscript. More scientific techno-economic analysis and life cycle assessment should be implemented in future work.

(2) The generation number for the ALE process can be easily estimated based on the OD change in each growth transfer. For example, if the OD increased from 0.5 to 15, that's a number close to 2^5 , then it is about 5 generations. For a total of 40 transfers with each having similar growth period, then a total of $5 \times 40 = 80$ generations was estimated. Using the number of transfers, although is a common practice, but is not a scientifically meaningful. It would be nice to have the estimated generations included.

Response: Thanks for your kindly suggestion. As the reviewer mentioned, in each growth transfer, the initial OD₆₀₀ was 0.5, and after 36-48 h cultivation, the OD₆₀₀ reached to 5.0. The total generation of 40 transfers is about 120 generations. The generation number for the ALE process has been estimated and added in the manuscript

(Page 6, Line 155).

(3) In the response to the comment #13, the authors indicated that "SA/OD levels of Hi-SA2 strain in flask, 5-L, and 50-L were 7.82, 2.45, and 2.78, respectively. It seems that in the scale-up fermentation, the biomass production is higher, therefore the SA production capacity per unit biomass in the scale-up fermentation is decreasing". It would be nice to include this in the discussion to show the potential of further improvement opportunity via fermentation optimization and scale-up in future.

Response: Thanks for your kindly suggestion. This sentence has been included in the discussion of manuscript (Page 16, Line 396-400).

Reviewer #2 (Remarks to the Author):

The manuscript has been carefully revised. Most comments have been addressed carefully. However, there are minor points:

Response: Thank you for your time and effort in improving the quality of our manuscript.

1. Please mention in the text that PGC91-rTE1-1 derivatives harboring additional TbFrd overexpression no longer accumulated by-product malate.

Response: Thanks for your comments. This information has been mentioned in the text of manuscript (Page 11, Line 259).

2. The experiment with mCherry localization is not informative. Based on the results, the only conclusion that the authors can have is that “the fluorescence distribution of MTS-mCherry was different from that of cytosolic mCherry”. This does not support the statement in the text “The fluorescence distribution of MTS-mCherry was consistent with mitochondrial localization, while mCherry without the signal peptide showed a cytosolic distribution (Fig. 4b).” I would suggest either remove all information about mCherry targeting from the manuscript or move it to Supplementary Materials with correct conclusions.

Response: To determine the distribution of the signal peptide, we recently fused the mitochondrial targeting sequence (MTS) of cytochrome c oxidase 5b to another green fluorescent protein, sfGFP (Superfolder GFP). MTS-sfGFP was successfully expressed in *Y. lipolytica*, and its fluorescence distribution was consistent with the mitochondrial localization by staining Mito-Tracker Red CMXRos. These results were updated in the Fig.4b of revised manuscript. We thank you again for your valuable suggestion.

Fig. 4 Subcellular localization of fumarate reductase. a. Schematic representation of

the mitochondrial localization of the reductive TCA cycle; **b.** Functional validation of the mitochondrial targeting sequence from cytochrome c oxidase 5b. Fluorescence microscopy of yeast strains expressing sfGFP and strained with Mito-Tracker Red CMXRos. Green fluorescent protein sfGFP with or without mitochondrial targeting sequence were expressed in Po1f strain, respectively; **c** Expression of cytosolic and mitochondrial soluble Frd in the evolved strain PGC91-rTE1-1 to enhance SA production. The initial concentration of glucose was 60 g/L. Data are presented as mean \pm s.e.m. (n = 3 biologically independent samples).

3. Line 68: repetition of the noun cycle.

Response: Thanks for your reminding, this error has been corrected.

Reviewer #3 (Remarks to the Author):

The authors have modified the manuscript carefully. I believe it is ready for publication.

Response: Thank you for your time and effort in improving the quality of our manuscript.

Reviewers' Comments:

Reviewer #1:

Remarks to the Author:

The authors have made the revisions to address the concerns raised by Reviewer1 in the specific comments.

Reviewer #2:

Remarks to the Author:

The manuscript has been carefully revised. All the comments have been addressed carefully.

We thank all the reviewers again for their thoughtful comments. For the additional editorial requests, we have carefully addressed them as indicated in the author checklist. All changes are highlighted in red in the revised manuscript file.

Reviewer #1 (Remarks to the Author):

The authors have made the revisions to address the concerns raised by Reviewer1 in the specific comments.

Response: Thank you for your time and effort in improving the quality of our manuscript.

Reviewer #2 (Remarks to the Author):

The manuscript has been carefully revised. All the comments have been addressed carefully.

Response: Thank you for your time and effort in improving the quality of our manuscript.